# Ranking Free RAG: Replacing Re-ranking with Selection in RAG for Sensitive Domains

## Abstract

In sensitive domains, Retrieval-Augmented Generation (RAG) must be interpretable and robust because errors here don't just mislead; they invite lawsuits, undermine scholarly credibility, and breach compliance. Stakeholders require traceable evidence, clear rationale for why specific evidence were selected, and safeguards against poisoned or misleading content. Yet current RAG pipelines use similarity-based retrieval with arbitrary top-$k$ cutoffs, offering no explanation for their selections, and remain vulnerable to data poisoning attacks. We propose `METEORA`, which replaces these existing drawbacks in RAG pipelines with rationale-driven selection; explicit reasoning that simultaneously guides evidence choice, explains decisions, and remains robust to RAG poisoning. `METEORA` operates in three stages. First, a general-purpose Large Language Model (LLM) is preference-tuned to generate rationales conditioned on the input query using direct preference optimization. Second, these rationales guide the **Evidence Chunk Selection Engine**, which employs a two-step process: (Step 1) pairing individual rationales with retrieved evidence for query-specific relevance and applying elbow detection (identifying sharp drops in similarity scores) to determine an adaptive cutoff point that eliminates the need for top-$k$ heuristics, thereby acquiring dataset-specific relevance, and (Step 2) optionally performing *context expansion* by adding neighboring evidence. Lastly, the rationales are used by a **Verifier LLM** to detect and filter poisoned or misleading evidence for *safe* generation. The framework provides explainable and interpretable evidence flow by using rationales consistently across both selection and verification. Our evaluation across six datasets shows `METEORA` delivers breakthrough performance on three fronts: (i) it achieves **13.41%** higher recall, and its `METEORA` w/o Expansion variant achieves **21.05%** higher precision than the best-performing baseline; (ii) it reduces the amount of evidence required to reach comparable recall by **80%**, which directly improves downstream answer generation accuracy by **33.34%**; and (iii) it strengthens adversarial defense, increasing F1 **from 0.10 to 0.44** and making RAG systems more resilient to poisoning attacks. The code is available in the anonymous GitHub repository[1].

## 1 Introduction

Traditional Retrieval-Augmented Generation (RAG) systems retrieve and re-rank thousands of document evidences to answer queries, mostly focusing on improving accuracy. They provide no explanation for ***why specific evidence*** was chosen over alternatives. This opacity creates critical problems in sensitive domains where understanding the reasoning behind AI decisions is not optional, rather it is essential for auditability and transparency. Consider a legal AI system analyzing merger agreements or a healthcare platform processing patient data. When these systems select certain evidence chunks to generate responses, stakeholders need to understand the selection rationale. Current similarity-based re-ranking methods provide only opaque similarity scores and rely on arbitrary top-$k$ cutoffs, offering no insight into the decision-making process (Glass et al., 2022; Reimers and Gurevych, 2019a; Devlin et al., 2019). This black box nature has become untenable as organizations deploy RAG

---

[1] `https://anonymous.4open.science/r/METEORA-DC46/README.md`

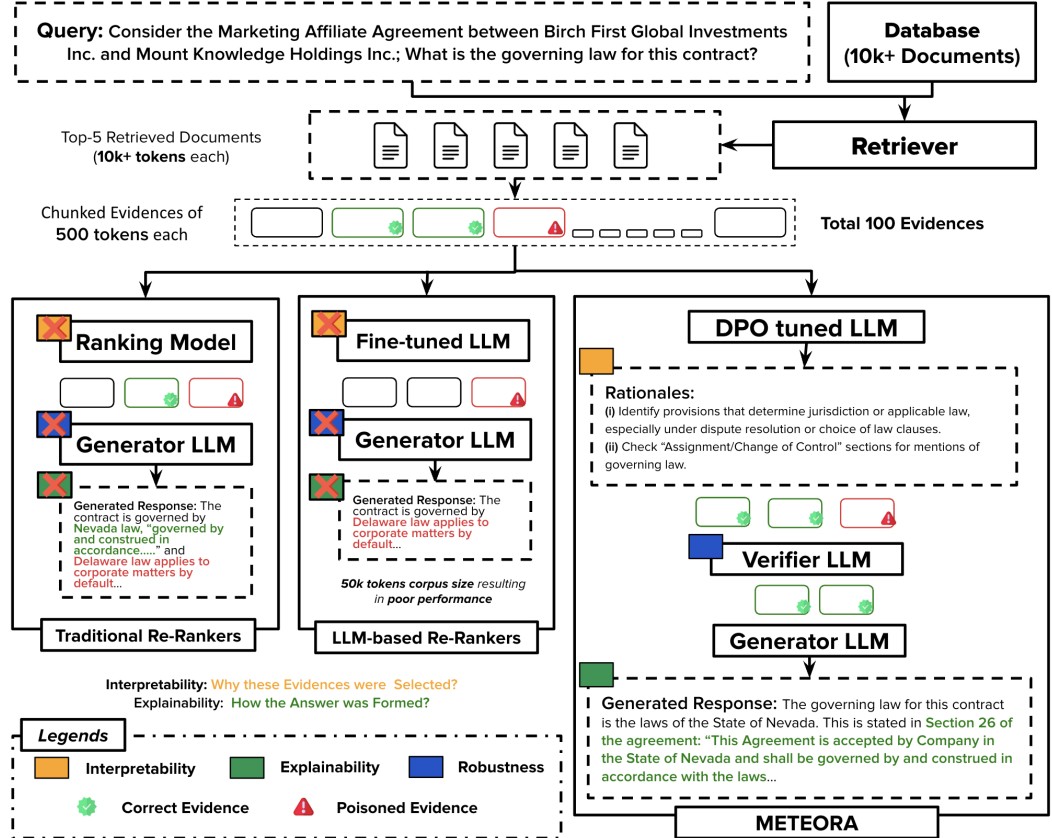

Figure 1: **METEORA achieves interpretable and robust evidence selection where existing approaches fail.** On a challenging legal query from Merger Agreement Understanding Dataset with poisoned evidence, traditional re-rankers select contaminated content producing factually incorrect answers, while LLM-based re-rankers fail due to context length limitations. **Only METEORA uses interpretable rationales to explain evidence selection and robustness with verification to filter poisoned content, achieving both interpretability and correctness.**

in high-stakes scenarios where wrong answers can have serious legal, financial, or medical consequences (Zhou et al., 2024a; Barron et al., 2024a).

The interpretability crisis becomes even more critical when considering adversarial threats. Recent demonstrations have shown how attackers can poison RAG knowledge bases to manipulate system outputs (Zou et al., 2024; Nazary et al., 2025; Chaudhari et al., 2024), but current re-ranking approaches provide no mechanism to detect such manipulation. Without understanding ***why evidence was selected***, organizations cannot audit whether the selection process has been compromised. The same opacity that prevents explainability also prevents robust defense against corpus poisoning attacks.

This challenge has intensified as RAG deployment has surged with enterprise adoption sky-rocketing to 50% in 2024, up from 31% the previous year (Research, 2024a). Simultaneously, researchers have demonstrated successful adversarial attacks against major RAG systems, including data poisoning exploits targeting ChatGPT's memory features (Research, 2024b) and systematic corruption of enterprise knowledge bases (SplxAI, 2024). In high-stakes domains, these adversarial attacks can have serious consequences. These attacks exploit the black box nature of evidence selection: if the system cannot explain its reasoning, it cannot detect when that reasoning has been subverted.

Current approaches miss the connection between interpretability and robustness. RAG$^2$ (Sohn et al., 2025a) and RADIO (Jia et al., 2025) attempt to improve retrieval through rationales but still rely on traditional re-ranking, inheriting its opacity. RankRAG (Yu et al., 2024a)

uses Large Language Models (LLMs) for ranking but provides no reasoning transparency and lacks mechanisms to detect adversarial manipulation. As illustrated in Figure 1, these methods treat interpretability and security as separate concerns: traditional re-rankers select irrelevant and poisoned evidence based on opaque similarity scores, while LLM-based re-rankers fail when context length exceeds their capacity, both lacking mechanisms to explain or verify their selection decisions.

We propose METEORA (Method for Interpretable rank-free evidence selection with Optimal Rationale), a framework that solves the black box problem by replacing opaque re-ranking with explicit reasoning. As demonstrated in Figure 1, METEORA generates rationales that explain *why* specific evidence is relevant to a query, then uses these same rationales to verify evidence consistency and detect adversarial content. Unlike LLM-based re-rankers that fail with long documents due to context length limitations, METEORA operates

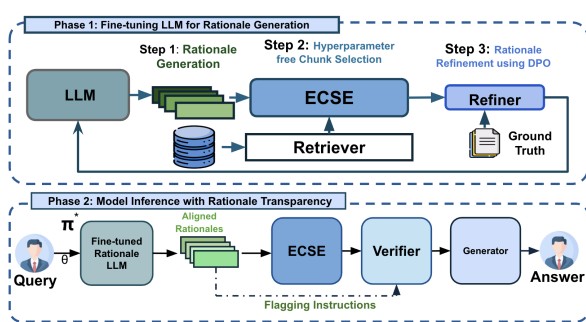

Figure 2: Overview of our METEORA framework.

without such constraints while providing a verifier that successfully filters poisoned evidence. Crucially, this enhanced interpretability and robustness comes ***without sacrificing efficiency***, METEORA's adaptive selection actually reduces computational overhead by processing only necessary evidence rather than arbitrary top-$k$ evidences.

Our key insight is that interpretable evidence selection, adversarial robustness, and computational efficiency are not competing goals but synergistic ones. By making the selection process explainable through rationales, we enable both human understanding and automated verification while simultaneously eliminating the computational waste inherent in arbitrary top-$k$ approaches. The same reasoning that justifies evidence selection can flag inconsistencies that indicate adversarial manipulation, and the adaptive nature of rationale-driven selection naturally processes only the evidence needed for accurate responses.

**Our main contributions are:**

- **DPO-based rationale generation framework:** To address the fundamental lack of explainability in RAG evidence selection, we introduce a novel approach for training rationale generators using Direct Preference Optimization (DPO) without manual annotation, enabling scalable production of query-aligned reasoning that makes evidence selection auditable and transparent in sensitive domains.
- **Evidence Chunk Selection Engine (ECSE) with statistical adaptive thresholding:** To eliminate the arbitrary and computationally wasteful nature of top-$k$ heuristics that process irrelevant evidence in current re-ranking methods, we develop an unsupervised evidence selection algorithm using statistical elbow detection with z-score normalization, providing principled, data-driven cutoff determination that achieves both computational efficiency and query-adaptive precision.
- **Unified rationale-based selection and verification:** To solve the fundamental disconnect between interpretability and adversarial robustness in current RAG systems, we establish a methodological framework where the same rationales that explain evidence selection also enable detection of corpus poisoning attacks, proving that transparency and security are not competing goals but synergistic properties of reasoning-based selection.

## 2 METEORA

**Problem Formulation:**

Formally, given a query $q$, knowledge base $D$, and retrieved documents $D_s \subset D$ chunked into evidence set $E$, the objective is to learn a rationale-based selection function $f_\theta(q, E) \rightarrow (R, E_s)$ that generates rationales $R = \{r_1, ..., r_k\}$ and selects evidence subset $E_s \subset E$ such that: (i) $E_s$ maximizes recall $R(E_s, E^*)$ where $E^*$ are ground-truth relevant evidences, (ii) $E_s$

minimizes precision loss from adversarial content $\mathcal{A} \subset E$, (iii) $|E_s|$ is determined adaptively without fixed top-$k$ constraints, and (iv) rationales $R$ provide interpretable justification for each $e \in E_s$. Figure 3 shows an example of a rationale generated for a real-world query from the PrivacyQA dataset.

## 2.1 Preference-Tuned Rationale Generator

`METEORA`'s rationale generator produces query-aligned reasoning that explains evidence selection decisions. Rather than relying on manual annotation for rationale creation (Wei et al., 2022; Kim et al., 2023; Zaidan et al., 2007), we use Direct Preference Optimization (DPO) to automatically train a general-purpose LLM to generate rationales that lead to correct evidence selection. DPO offers significant advantages over traditional RLHF approaches: it eliminates the need for reward model training and complex reinforcement learning pipelines, provides more stable training through simple classification loss, and requires substantially lower computational resources (Rafailov et al., 2023a).

**Preference Dataset Construction.** We automatically construct preference datasets from existing QA annotations without manual labeling. For each query $q$ and ground-truth evidence $e^*$, we generate multiple rationales and label those that lead to correct evidence selection as preferred ($r_w$) and those that select incorrect evidence as dispreferred ($r_l$). This approach leverages the insight that rationale quality can be measured by selection accuracy rather than human judgment.

**DPO Training Objective.** We train the rationale generator using the DPO loss function:

$$\mathcal{L}_{DPO}(\pi_\theta; \pi_{\text{ref}}) = -\mathbb{E}_{(q,e,r_w,r_l)\sim\mathcal{D}} \left[ \log \sigma \left( \beta \log \frac{\pi_\theta(r_w|q,e)}{\pi_{\text{ref}}(r_w|q,e)} - \beta \log \frac{\pi_\theta(r_l|q,e)}{\pi_{\text{ref}}(r_l|q,e)} \right) \right] \quad (1)$$

where $\pi_{\text{ref}}$ is the frozen reference model, $\pi_\theta$ is the trainable model with the same architecture, and $\beta$ controls preference learning intensity. The model learns to assign higher likelihood to rationales $r_w$ that lead to correct evidence selection over rationales $r_l$ that select incorrect evidence, given query $q$ and evidence $e$.

**Training vs. Inference.** During training, we condition on both query and ground-truth evidence: $\pi_\theta(r|q, e^*)$, enabling the model to learn what good rationales look like when paired with correct evidence. At inference time, we condition only on the query: $\pi_\theta(r|q)$, allowing the model to generate query-appropriate rationales that guide evidence selection. This training paradigm enables the model to internalize effective reasoning patterns and apply them to new queries, achieving superior performance compared to baselines (section 4).

The resulting preference-tuned model $\pi_\theta^*$ generates query-aligned rationales that serve dual purposes in `METEORA`: guiding evidence selection through the Evidence Chunk Selection Engine (subsection 2.2) and enabling adversarial detection through the Verifier LLM (subsection 2.3). Implementation details are provided in Table 5, with prompts in §A.3 and qualitative examples in §A.2.

**Note on Adversarial Robustness.** The preference-tuned rationale generator provides implicit defense against coincidental selection of adversarial evidence through three

> **Example**
>
> **Query:** TickTick: To Do List with Reminder, Day Planner's privacy policy; can it view my real name?
>
> **Rationale:** Search for terms like `"real name"`, `"PII"`, or `"user information"`, especially in sections covering data collection, use, or disclosure. *Flagging Instructions:* Flag the evidence if it contains internally inconsistent language about real name usage, or if it contradicts other verified parts of the policy.

Figure 3: Generated Rationale

mechanisms: (1) specificity learning, where DPO training optimizes rationales for fine-grained features that distinguish correct from incorrect evidence, making coincidental matching with adversarial content less likely; (2) query-evidence alignment, where conditioning on $\pi_\theta(r|q, e^*)$ learns specific relational patterns that adversarial content would need to coincidentally match; and (3) selection accuracy pressure, where the preference objective reinforces rationales robust to superficial similarities. However, this implicit defense has limitations; negative examples $r_l$ represent naturally incorrect evidence rather than adversarially crafted content, creating a training distribution gap. While rationale specificity provides some protection, the

primary defense against adversarial evidence comes from the explicit consistency checking performed by the Verifier LLM (subsection 2.3), which uses the same rationales for flagging inconsistent or malicious content.

## 2.2 ECSE: Unsupervised Rationale-Based Evidence Selector

The Evidence Chunk Selection Engine (ECSE) is the core component that **eliminates top-$k$ heuristics** by using generated rationales to adaptively determine how many evidence chunks to select. ECSE operates through two complementary mechanisms that collectively identify relevant evidence without relying on arbitrary cutoffs.

**Stage 1: Pairing Rationales with Evidence.** The DPO-tuned LLM generates *locally relevant, query-specific rationales*, meaning each rationale is tailored to and depends solely on the individual query being processed. To capture this query-specific trait, we perform rationale-based pairing. Each rationale captures a specific aspect of what makes evidence relevant to the query. To leverage this specificity, we pair each rationale $r_i \in \mathcal{R}$ with its most semantically similar evidence chunk (by cosine similarity), forming the set $E_v$:

$E_v = \left\{ \arg\max_{e_j \in E} \ \mathcal{S}(r_i, e_j) \mid r_i \in \mathcal{R} \right\}$, where $\mathcal{S}(\cdot, \cdot)$ denotes cosine similarity.

*Note on Rationale Convergence:* Multiple rationales may select the same evidence chunk when they capture complementary aspects of the same relevant content, resulting in $|E_v| \leq |\mathcal{R}|$. This convergence is desirable as it indicates consensus among rationales about highly relevant evidence. Rather than forcing diversity, we interpret this as evidence validation, when multiple reasoning paths point to the same content, it strengthens confidence in that evidence's relevance. This pairing ensures that each rationale's specific intent is represented in the selected evidence, providing high precision through rationale-evidence alignment.

*Global Relevance/Dataset-Specific:* Further, to capture the dataset-specific trait, we identify evidence that align with the overall intent of all rationales. To identify such evidence, we compute a pooled rationale embedding: $\bar{r} = \frac{1}{|\mathcal{R}|} \sum_{r_i \in \mathcal{R}} \text{SBERT}(r_i)$

We rank all evidence chunks by similarity to $\bar{r}$, yielding similarity scores $\{s_1, s_2, \ldots, s_n\}$ in descending order. To determine the adaptive cutoff $k^*$ without top-$k$ heuristics, we apply statistical elbow detection: (a) Compute first-order differences: $\Delta_i = s_i - s_{i+1}$; (b) Apply z-score normalization: $z_i = \frac{\Delta_i - \mu_\Delta}{\sigma_\Delta}$ where $\mu_\Delta$ and $\sigma_\Delta$ are the mean and standard deviation of all differences. (c) The selection index $k^*$ is identified at the first point where the drop in similarity significantly deviates from the average pattern, indicating a natural boundary between highly relevant and less relevant evidences.

This identifies the point where similarity drops significantly, indicating a natural boundary between relevant and irrelevant evidence. If no significant drop is detected (all $z_i \leq \tau$), we use second-order differences $\nabla_i^2 = \Delta_{i+1} - \Delta_i$ to find maximum curvature: $k^* = \arg\max_i |\nabla_i^2|$. The selected evidence set is $E_g = \{e_1, e_2, \ldots, e_{k^*}\}$.

**Stage 2: Context Expansion (Addressing Chunking Brittleness).** Document chunking can artificially split coherent information across adjacent evidences. To recover potentially fragmented context, we expand each selected evidence in $E_v \cup E_g$ by including its immediate neighbors: $E_w = \{e_{j-1}, e_{j+1} \mid e_j \in (E_v \cup E_g), j - 1 \geq 1, j + 1 \leq |E|\}$

The final evidence set is $\mathbf{E_s} = E_v \cup E_g \cup E_w$. Note that this step is optional; if the chunking process is sufficiently robust, it can be skipped (Stäbler et al., 2025). The expansion involves a precision-recall trade-off: while it recovers split information, it may introduce irrelevant content when evidences are not actually fragmented.

**Computational Complexity Analysis.** To analyze METEORA's efficiency, we compare the computational costs of traditional RAG against our approach using standard big-O notation, where $n$ represents the number of retrieved evidences, $r$ the number of rationales, and $d$ the embedding dimension. Traditional RAG systems follow a straightforward three-step process. Re-ranking requires $O(n \cdot d)$ operations to compute similarity scores between the query and all $n$ evidences. Top-$k$ selection then sorts these evidences in $O(n \log n)$ time to identify the most relevant ones. Finally, generation processes the selected $k$ evidences through the LLM at cost $O(k \cdot L_{\text{prefill}})$, where $L_{\text{prefill}}$ represents the computational expense of LLM processing per evidence.

METEORA's pipeline involves more steps but achieves better overall efficiency. Rationale generation requires $O(L_{\text{rat}})$ for a single LLM call. ECSE pairing then computes similarities between $r$ rationales and all $n$ evidences at cost $O(r \cdot n \cdot d)$. The pooling stage combines rationale embeddings in $O(r \cdot d)$ time, computes evidence similarities in $O(n \cdot d)$, sorts results in $O(n \log n)$, and performs z-score elbow detection in $O(n)$, totaling $O(r \cdot d + n \cdot d + n \log n + n)$. Context expansion adds $O(|E_v \cup E_g|)$ operations for neighbor lookup. Finally, generation processes only $k^*$ adaptively selected evidences at cost $O(k^* \cdot L_{\text{prefill}})$. METEORA selects about $\sim$**80% fewer evidences** than the traditional top-$k$ baseline for achieving similar recall, that is $k^* \ll k$. With short generations, $L_{\text{prefill}} \gg L_{\text{rat}}$, so prefill dominates total cost (Agrawal et al., 2024). Since both methods produce outputs of similar length, reducing the context size directly reduces compute, yielding substantial net savings for METEORA.

## 2.3 Verifier LLM

To enhance METEORA's robustness against obvious contradictions and factual inconsistencies, we incorporate a Verifier LLM that performs conservative consistency checking on selected evidence $E_s$ before generation. The Verifier uses the same rationale framework that guides evidence selection, providing a secondary filtering mechanism within METEORA's unified reasoning approach.

**Verification Process.** The Verifier evaluates each evidence using the input query, associated rationales (which serve as flagging instructions), and summaries of previously verified evidences. Following a conservative design philosophy[2], the Verifier assumes validity unless strong evidence suggests otherwise and only flags content when highly confident ($> 90\%$). Evidence is flagged for three types of issues: (1) *factual violations*, when content contradicts well-established facts with high confidence; (2) *contradiction*, when evidence is logically inconsistent with previously verified evidences; and (3) *instruction violations*, when evidence fails to meet criteria embedded in the rationales (Greshake et al., 2023; Yi et al., 2023; Zou et al., 2024; Clop et al., 2024; Ben-Tov et al., 2025; Parry et al., 2024). Flagged evidence is discarded, and only the filtered set proceeds to generation

**Implementation.** We use the same `Llama-3.1-8b-instruct` model for verification to maintain consistency across the framework. The Verifier processes each evidence independently and outputs structured decisions including flagging status, evidence summaries, and violation types. This conservative approach prioritizes precision over recall in adversarial detection, reducing false positives that could harm system performance on clean data.[3] For more information about the prompt format, see §A.3.

## 3 Experiments

To demonstrate the effectiveness of rationales, we evaluate METEORA on three tasks across six real-world benchmark datasets spanning multiple domains, and provide a detailed ablation study. For clarity, we report results using evidence of chunk size of 512 tokens in the main paper; results for other sizes are included in §A.6.

**Tasks and Datasets:** We evaluate METEORA's core properties through three complementary tasks that measure different aspects of rationale-driven evidence selection.

**(a)** *Context Prioritization (CP)* measures METEORA's fundamental capability to eliminate top-$k$ heuristics through adaptive evidence selection. This task evaluates precision, recall, and F1 scores against human-annotated ground truth evidence spans, directly testing whether statistical elbow detection identifies relevant content more accurately than arbitrary cutoffs. We used `LLaMA-3.1-8b-Instruct` for rationale generation, evidence verification,

---

[2]**Design Limitations.** The Verifier operates within the same rationale-based reasoning framework used for evidence selection, creating a potential single point of failure. If adversarial content is crafted to exploit the specific rationale patterns learned during DPO training, both selection and verification mechanisms could be compromised simultaneously. We design the Verifier as a conservative consistency checker rather than a comprehensive adversarial defense system.

[3]We consider adversarial attacks targeting the rationale generation model itself (e.g., model poisoning, training data manipulation) to be outside the scope of this work. Our focus is on detecting inconsistencies in retrieved content rather than defending against compromised reasoning frameworks.

and answer generation. Baseline re-ranking methods varied top-k from 1-64 across datasets and evidence sizes. Since `METEORA` doesn't use fixed k-values, for fair comparison, we set each baseline's k-value to match the average number of evidences selected by `METEORA` and measure performance using Precision@k and Recall@k.

**(b)** *Generation Quality* assesses whether improved evidence selection translates to better answer accuracy. By comparing generated responses against reference answers, this task validates that selecting fewer but more relevant evidences enhances downstream performance, the core efficiency claim of adaptive selection. We measure generation quality using `GPT-4o` as a binary evaluator (+1 correct, 0 incorrect).

**(c)** *Adversarial Defense* Adversarial Defense: Following Nazary et al. (2025)'s protocol, we simulated corpus poisoning using domain-specific LLMs to generate semantically coherent but factually incorrect content. We used `Law-Chat` for legal datasets, `Finma-7b-full` for financial data, and `LLaMA-3.1-8b-Instruct` for academic content. We randomly poisoned 30% of QA instances by inserting malicious text into documents containing correct context. This creates a particularly challenging detection scenario since poisoned content appears alongside legitimate information in the same documents, making it difficult to distinguish based on location or semantic similarity alone. Adversarial robustness was measured using precision and recall for detecting poisoned content.

**Dataset Selection Criteria.** We selected six datasets spanning legal, financial, and academic domains based on three evaluation requirements: (1) expert-annotated evidence spans for precise CP evaluation, (2) lengthy documents (5-50 pages) that stress-test adaptive selection under high ev-

Table 1: Dataset Statistics Across Domains

| Dataset | Docs | Avg. Tokens | QA Pairs | Domain |
|---|---|---|---|---|
| ContractNLI | 95 | 10,673 | 946 | Legal (NDA) |
| CUAD | 462 | 55,827 | 4,042 | Legal (Contracts) |
| MAUD | 150 | 351,476 | 1,676 | Legal (M&A) |
| PrivacyQA | 7 | 25,266 | 194 | Legal (Privacy) |
| FinQA | 2,789 | ~700 | 8,281 | Finance |
| QASPER | 1,585 | ~6,500 | 5,049 | Academic (NLP) |

idence counts, and (3) domain diversity to validate generalization without parameter retuning. Legal datasets include `ContractNLI` (Koreeda and Manning, 2021), `PrivacyQA` (Ravichander et al., 2019), `CUAD`(Hendrycks et al., 2021), and `MAUD` (Pipitone and Alami, 2024), representing increasing complexity from straightforward privacy policies to highly technical merger and acquisition agreements (M&A). `FinQA` provides numerical reasoning challenges over financial reports with hybrid text-table evidence (Chen et al., 2021). `QASPER` covers academic research papers where questions target specific methodological or empirical claims requiring precise evidence selection (Dasigi et al., 2021).Table 1 shows the statistics. This evaluation framework directly tests `METEORA`'s core contributions: interpretable evidence selection through explicit rationales, principled elimination of arbitrary top-$k$ heuristics via adaptive thresholding, and robustness against adversarial content through rationale-based verification.

**Baselines.** We evaluate against established re-ranking methods representing different paradigms in retrieval systems. For context prioritization, we compare with Cross-Encoder (ms-marco-MiniLM-L4-v2, bge-reranker-large) (Huggingface), which processes query-document pairs jointly to produce fine-grained relevance scores. We also test against Contriever (Izacard et al., 2022), an unsupervised dense retriever that learns representations through contrastive learning, and SBERT (ms-marco-MiniLM-L4-v2, e5-large-v2) (Reimers and Gurevych, 2019b), which computes similarity between independent query and document embeddings. Additionally, we include Fine-Tuned SBERT (Legal-huggingface), a domain-adapted version specialized for legal text, to assess the impact of domain-specific training. For LLM-based comparison, we use LLM-based rerankers such as RankLLaMa (Ma et al., 2024) and implement state-of-the-art RankRAG using the Promptriever model (Weller et al., 2024), which leverages large language models to directly rank and select relevant evidences. Finally, we evaluate adversarial robustness against Perplexity-based Defense (Zhou et al., 2024b), which detects poisoned content by measuring deviations from expected language model behavior.

## 4 RESULTS

**Results from CP and Generation Tasks:** Table 2 reports performance on the CP task across all datasets; our baselines include six traditional re-rankers and two LLM-based re-rankers; the `METEORA` framework not only provides interpretability (see Figure 1 and

Table 2: CP Task Results across Datasets: METEORA achieves the best average recall and precision, outperforming all baselines. Since METEORA does not depend on a specific K value, we established a fair comparison by taking the average number of evidences it selected and using that number as the K value for all other baselines. The metrics are Precision (P@K) and Recall (R@K). Values in **bold** exceed state-of-the-art alternatives.

| Model | QASPER | | Contract-NLI | | FinQA | | PrivacyQA | | CUAD | | MAUD | | Average | |
|---|---|---|---|---|---|---|---|---|---|---|---|---|---|---|
| | R@8 | P@8 | R@3 | P@3 | R@10 | P@10 | R@6 | P@6 | R@12 | P@12 | R@33 | P@33 | R | P |
| SBERT (MiniLM-L4-v2) | 0.91 | 0.26 | 0.91 | 0.38 | 0.96 | 0.12 | 0.89 | 0.24 | 0.78 | 0.11 | 0.41 | 0.03 | 0.81 | 0.19 |
| SBERT (E5-Large-v2) | 0.94 | 0.27 | 0.92 | 0.34 | 0.96 | 0.12 | 0.78 | 0.21 | 0.77 | 0.12 | 0.44 | 0.02 | 0.80 | 0.18 |
| Contriever | 0.94 | 0.25 | 0.89 | 0.36 | 0.98 | 0.10 | 0.82 | 0.23 | 0.73 | 0.10 | 0.46 | 0.01 | 0.80 | 0.17 |
| Cross-Encoder (MiniLM-L4-v2) | 0.94 | 0.27 | 0.91 | 0.38 | 0.97 | 0.10 | 0.81 | 0.22 | 0.71 | 0.10 | 0.50 | 0.02 | 0.80 | 0.18 |
| Cross-Encoder (BGE-Reranker-Large) | 0.93 | 0.27 | 0.92 | 0.37 | 0.97 | 0.11 | 0.85 | 0.22 | 0.78 | 0.11 | 0.51 | 0.02 | 0.82 | 0.18 |
| Finetuned-SBERT | 0.92 | 0.26 | 0.92 | 0.39 | 0.97 | 0.13 | 0.75 | 0.21 | 0.76 | 0.11 | 0.46 | 0.02 | 0.79 | 0.18 |
| RankLLaMA (7b) | 0.75 | 0.19 | 0.73 | 0.24 | 0.85 | 0.08 | 0.83 | 0.15 | 0.71 | 0.08 | 0.27 | 0.02 | 0.69 | 0.12 |
| RankRAG (8b) | 0.76 | 0.19 | 0.77 | 0.23 | 0.89 | 0.07 | 0.86 | 0.19 | 0.60 | 0.07 | 0.22 | 0.01 | 0.68 | 0.13 |
| METEORA w/o DPO | **0.96** | 0.25 | **0.95** | 0.37 | 0.92 | 0.11 | **0.92** | 0.22 | **0.89** | 0.12 | **0.65** | 0.02 | **0.88** | 0.18 |
| METEORA | **0.99** | 0.26 | **1.00** | 0.35 | 0.95 | 0.12 | **0.98** | 0.23 | **0.93** | 0.12 | **0.72** | 0.03 | **0.93** | 0.19 |
| METEORA w/o Expansion | **0.96** | **0.31** | **0.97** | **0.45** | 0.94 | **0.14** | **0.96** | **0.27** | **0.90** | **0.14** | **0.66** | **0.04** | **0.89** | **0.23** |

§A.2) but also surpasses state-of-the-art baselines in retrieval performance. On individual datasets, METEORA improves recall over the best-performing baseline by 5.31% (QASPER), 8.6% (Contract-NLI), 10.11% (PrivacyQA), 19.23% (CUAD), and 41.17% (MAUD); these results indicate that as dataset complexity and average document length increase from 6.5k to 350k tokens (see Table 1), the advantage of METEORA becomes more pronounced. METEORA's advantage grows as datasets become more challenging, precisely where similarity-based methods hit their ceiling: **(a)** DPO training first learns to bridge the gap between vague questions and where answers actually reside, aligning intent with evidence-bearing regions (for a better theoretical understanding, see §D), in Scientific QASPER, where questions are written from titles and abstracts while answers lie deep in full papers, and transferring the same behavior to legal corpora, where it maps queries to the specific clauses that support, contradict, or ignore a claim. **(b)** Rationale generation then converts this learned intent into precise retrieval targets, turning "How does this model work?" into searches for methodology sections, experimental protocols, and evaluation metrics in scientific papers, and showing the parallel shift in legal datasets from surface keywords in Contract-NLI to higher level concepts in CUAD and MAUD, recognizing that "liability limitation," "damages exclusion," and "loss indemnification" refer to same concept. **(c)** Adaptive thresholding finally decides how much evidence to gather, adjusting to uneven information density in both scientific articles and contracts, stopping when sufficient relevant content is found; simple papers or contracts need fewer chunks, complex studies or merger agreements need more, and the statistical detector finds natural breakpoints without fixed cutoffs.

On FinQA, which has relatively short documents (average length ~700 tokens), METEORA is not the most effective method in terms of recall. This is because FinQA is constructed from passages that already contain the answer, rather than from entire financial documents. Such a setup does not reflect real-world scenarios, where the answer could appear anywhere in a long, complex document. In this setting, traditional similarity-based re-rankers achieve higher recall (0.98 vs. 0.95). Nevertheless, METEORA achieves a significant reduction in number of evidences (~**80% fewer evidence**) than strong baselines, which cuts noisy context and directly boosts generation accuracy with more grounded responses (see Figure 4).

Table 3: Baselines require proportionally more evidences than METEORA (values > 1×) to reach comparable recall, except in FinQA (values < 1×)

| Method | QASPER | C-NLI | FinQA | PrvQA | CUAD | MAUD | Avg |
|---|---|---|---|---|---|---|---|
| SBERT | 1.88× | 21.33× | 0.90× | 2.00× | 2.33× | 1.46× | 4.98× |
| Cross-Encoder | 1.75× | 21.33× | 0.80× | 2.17× | 2.17× | 1.70× | 4.99× |
| Contriever | 1.88× | 21.33× | 0.90× | 2.00× | 2.42× | 1.82× | 5.06× |
| RankRAG | 4.00× | 21.33× | 1.50× | 3.33× | 5.17× | 3.15× | 6.41× |

**Robustness to Corpus Poisoning in RAG:** METEORA successfully defends against attacks and produces correct answers, as shown in Table 4. Without defense, the system fails completely (F1 = 0), and basic perplexity checks provide little help. Most caught poisoned evidence gets flagged for *instruction violations* rather than contradictions or factual errors. This shows that attacks mainly try to change *how the model works* instead of changing facts. Since the Verifier only flags evidence when confident and removes it before generating answers, the improvements come from *selection*, not from adding text. Even with shorter evidence, brief rationale-based cues are enough to decide what to keep and what to remove.

**Ablation Effect of DPO Training.** DPO fine-tuning demonstrates its critical role in bridging the semantic gap between surface-level queries and the deep document locations where answers actually reside. The improvement is noticeable in complex datasets, such as MAUD experiences a 23.7% recall boost, while simpler datasets like FinQA show minimal gains (2.1%). This pattern reveals DPO's core mechanism; it teaches models to map abstract legal concepts to specific document structures. The training paradigm of conditioning on both query and

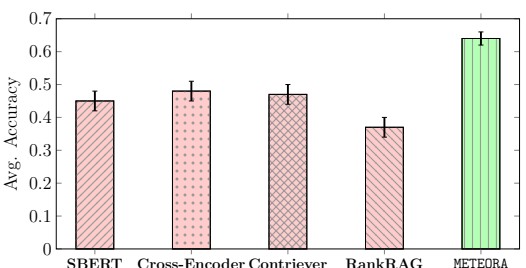

Figure 4: Response accuracy when retrieved evidence matches ground-truth, with variance bars indicating statistical significance (p < 0.01, paired t-test) across trials.

ground-truth evidence during training, then generating rationales from queries alone at inference, enables the model to internalize these complex document navigation patterns **without requiring explicit structural annotations**.

**Context Expansion Ablation.** The expansion stage reveals a nuanced relationship between document complexity and chunking brittleness that challenges conventional assumptions about context windowing. MAUD's merger agreements, with their intricate cross-references and nested clause structures averaging 351k tokens, benefit significantly from expansion (7% recall ↑) because related legal concepts span adjacent chunks due to complex interdependencies. Contracts in ContractNLI show minimal gains from context expansion because they utilize well-structured organizational patterns with clearly delineated sectional boundaries, where relevant information remains localized within individual chunks rather than distributed across adjacent segments. QASPER's academic papers demonstrate moderate benefits as methodology discussions often span multiple chunks, requiring adjacent context to maintain coherence . PrivacyQA shows substantial improvement because privacy policy statements frequently continue across artificially imposed chunk boundaries. This dataset-specific performance pattern validates our hypothesis that expansion effectiveness correlates with document structural over document size alone, suggesting adaptive expansion strategies could optimize the precision-recall tradeoff.

**Verifier Ablation.** Instruction-based flagging accounts for 87% of detected adversarial content, revealing that attackers primarily target procedural subversion rather than factual substitution. This validates our rationale-based verification approach: adversaries attempt to manipulate "how to search" rather than "what facts exist." Contradiction-based detection contributes minimally (below 2%), while factual violation detection provides supplementary

Table 4: Robust to corpus poisoning. **Top:** F1 score across datasets. **Bottom**: % of poisoned evidence flagged by category.

| Method | QASPER | C-NLI | FinQA | PrivQA | CUAD | MAUD |
|---|---|---|---|---|---|---|
| No Defense | 0.00 | 0.00 | 0.00 | 0.00 | 0.00 | 0.00 |
| Perplexity | 0.11 | 0.08 | 0.11 | 0.15 | 0.09 | 0.06 |
| METEORA | **0.44** | **0.51** | **0.43** | **0.48** | **0.33** | **0.39** |

| Flag Type | QASPER | C-NLI | FinQA | PrivQA | CUAD | MAUD |
|---|---|---|---|---|---|---|
| Instruction | **32.13** | **42.62** | **37.02** | **43.20** | **24.55** | **46.08** |
| Contradiction | 0.23 | 0.24 | 1.84 | 0.64 | 0.31 | 0.00 |
| Factual | 9.62 | 1.13 | 0.11 | 5.15 | 1.14 | 0.00 |
| Total | 41.98 | 43.99 | 38.97 | 48.99 | 26.00 | 46.08 |

value, particularly in QASPER (9.62%). The interpretable behavior of verifier prevents false positive cascades in RAG and maintain robust adversarial defense.

## 5 CONCLUSION

In this work, we presented METEORA, a rationale-driven, ranking-free framework specifically tailored for sensitive domains. We showed that by eliminating top-$k$ heuristics and opaque scoring functions, our approach reduces evidence requirements by 80% compared to traditional baselines. We found that rationale-based selection enables interpretability, robustness, and efficiency to work synergistically rather than compete. We conducted thorough validation across six datasets, demonstrating that METEORA consistently selects more relevant evidence while providing transparent evidence flow and strong resilience to adversarial content. In future work, investigating rationale generation for broader retrieval tasks and exploring the scalability limits of rationale-based selection across increasingly complex document types may yield further advances in interpretable retrieval systems.

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

# A    APPENDIX

## A.1    RELATED WORK

**RAG in Sensitive Domains.** RAG has been widely adopted in high-stakes domains such as law, finance, and healthcare, where factual accuracy and verifiability are critical (Lewis et al., 2021; Li et al., 2025; Sohn et al., 2025b; Karpati and Szabo, 2023; Barron et al., 2024b; Bhushan et al., 2025). Such domains are regulated and hence require traceable generations to retrieved and selected sources (for example, denial of loan application) and are prone to select semantically similar but contextually misleading evidence (Karpati and Szabo, 2023; Bhushan et al., 2025). Furthermore, RAG pipelines remain susceptible to corpus-poisoning attacks (Verma et al., 2024; Zhou et al., 2024b), highlighting the critical need for secure, context-aware retrieval methods. Our framework addresses these vulnerabilities by producing query-specific rationales that explain and justify chunk/evidence selection, thereby enhancing the system's transparency, interpretability, and overall robustness.

**Heuristics in RAG.** Most RAG systems use fixed top-$k$ selection heuristics, which often hurt performance due to irrelevant or noisy context (Yoran et al., 2024; Asai et al., 2023). Moreover, in real-world applications, it is hard to know upfront what the value should be for $k$, and re-rankers often lack interpretability, and hence complicates their deployment in sensitive domains. While some efforts have been made to automate the selection of $k$ (Chen et al., 2017; Ren et al., 2025), these methods had limited success due to their inability to explain the selection of evidence to generate final text. Dynamic retrieval methods such as RankRAG (Yu et al., 2024b) and Self-RAG (Asai et al., 2023) improve adaptability but lack interpretability. In contrast, our `METEORA` replaces top-$k$ heuristics with rationale-grounded selection, enabling query-specific and explaninable document filtering.

**Interpretability in RAG**. Interpretability is often absent in RAG pipelines. Recent efforts such as MIRAGE (Qi et al., 2024) and Rationale-first Retrieval (Sohn et al., 2025b) introduce rationales or attribution signals, but they focus on improving generation faithfulness and retriever training, respectively. In `METEORA`, we target the re-ranking stage and define end-to-end rationale integration by using rationales both to select evidence and to verify them before generation. Set-R (Lee et al., 2025) also employs rationales during re-ranking, but it continues to use LLM-based ranking and does not address adversarial robustness, which limits ranking and verification effectiveness compared with `METEORA`. Unlike Sohn et al. (2025b), which still relies on downstream re-ranking and lacks verification, `METEORA` removes re-ranking entirely and applies rationale-derived instructions in both selection and filtering, yielding interpretable decisions.

**Reliability in RAG.** RAG systems are susceptible to adversarial attacks and retrieval of noisy evidence (Verma et al., 2024; Xue et al., 2024). Methods like entailment filtering (Yoran et al., 2024) and ensemble retrieval offer partial solutions. These approaches address only specific vulnerabilities while leaving others unaddressed - entailment filtering can be overly strict and discard valuable information that doesn't meet formal entailment criteria, while ensemble methods add complexity and computational overhead without fundamentally solving the semantic verification problem. On the other hand, `METEORA` adds semantic filtering by requiring coherent rationales per evidence, to help discard poisoned evidence before generation.

**Feedback-Based Optimization.** Policy optimization methods like PPO (Ouyang et al., 2022), RLHF, and GRPO (Yang et al., 2023) have been used to align LLMs with human preferences, but rely on complex reward models and unstable training dynamics. In contrast, DPO (Rafailov et al., 2023b) offers a simpler, supervised alternative that directly optimizes preferred outputs. While prior methods focus on generation helpfulness or safety, we demonstrate that DPO is well-suited for rationale generation, where precision and domain adherence are crucial. PPO and RLHF suffer from costly reward modeling, opaque optimization processes, and vulnerability to data poisoning, whereas GRPO reduces but doesn't eliminate these issues. DPO overcomes these limitations by providing explicit, interpretable reasoning paths for evidence selection, making the rationale generation process not only more transparent but also more robust against poisoned content. We apply DPO in `METEORA` to produce rationales that align with domain-specific expectations while avoiding the pitfalls of traditional policy gradients.

## A.2   Qualitative Examples Showcasing METEORA's Interpretable and Explainable Nature

**Positive Example**

**Explainability: How the Answer Was Formed**

**Interpretability: Why This evidence Was Selected**

**Query:** *Consider the Marketing Affiliate Agreement between Birch First Global Investments Inc. and Mount Knowledge Holdings Inc.; What is the governing law for this contract?*

**Rationales:**
- Identify provisions that determine jurisdiction or applicable law, especially under dispute resolution or choice of law clauses.
- Check "Assignment/Change of Control" sections for mentions of governing law.

**Selected evidence:**
> *"This Agreement is accepted by Company in the State of Nevada and shall be governed by and construed in accordance with the laws thereof, which laws shall prevail in the event of any conflict."*

**Generated Answer:**
*The governing law for this contract is the laws of the State of Nevada. This is stated in Section 26 of the agreement: "This Agreement is accepted by Company in the State of Nevada and shall be governed by and construed in accordance with the laws…"*

**GPT-4o Evaluation:**
**Accuracy Score:** 1
**Analysis:** The answer correctly identifies the governing law and references the exact clause, showing clear alignment between rationale, context, and final output.

Figure 5: Positive example demonstrating how METEORA links rationale-based evidence selection (interpretability) with rationale-grounded answer generation (explainability), resulting in a correct and traceable response to a legal question.

---

**Negative Example**

**Explainability: Why the Answer Was Incorrect**

**Interpretability: Why These evidence Were Selected**

**Query:** *Consider the Marketing Affiliate Agreement between Birch First Global Investments Inc. and Mount Knowledge Holdings Inc.; Is there a cap on liability under this contract?*

**Rationales:**
- Look for sections discussing liability caps or clauses stating a maximum limit of responsibility for either party.
- Search for phrases like "shall not exceed," "limited to," or "capped at" in liability or indemnity sections.
- Focus on sections near "Dispute Resolution" or "Indemnification" — common locations for liability limitations.

**Correct evidence (not selected):**
> *"Company is not liable for incidental, special or consequential damages for any reason [...]. Company's liability shall not exceed the fees that MA has paid under this Agreement."*

**Generated Answer:**
*There is no clear cap on liability under this contract; however, according to paragraph 15, Company's liability "shall not exceed the fees..."*

**GPT-4o Evaluation:**
**Accuracy Score:** 0
**Analysis:** The generated answer inaccurately states that there is no clear cap on liability and does not fully convey the information that the company's liability is capped at the fees paid under the agreement. The reference answer provides a clear and specific cap on liability, which is missing in the generated answer.

Figure 6: Negative example showing how `METEORA` enables transparent error tracing. Although the correct evidence was not selected, the rationale trail provides insight into why the incorrect evidences were selected, and how they influenced the erroneous answer.

### A.3  PROMPTS USED IN EXPERIMENTS

**Few-Shot Prompt Template for Rationale Generation**

**Instructions:** Generate semantic rationales to guide query-specific information selection from domain-specific documents. Each rationale should:
- Represent a unique semantic search strategy grounded in the query.
- Be concise, concrete, and tailored to the scientific, legal, or financial context.
- Help extract precise and targeted evidence from long-form documents.
- Avoid redundancy across rationales.

**Formatting Guidelines:**
- Use XML-style tags: `<rationale_1>`, `<rationale_2>`, etc.
- Include a brief description in square brackets.
- Follow with a strategic, query-specific rationale sentence.

**Example Query:** *What are the limitations of this approach?*

**Example Response (truncated):**
```
<rationale_1>[Locate explicit limitation sections] Look for sections explicitly titled "Limitations,"
"Threats to Validity," or "Shortcomings" which directly enumerate the authors' acknowledged
limitations......</rationale_1>
...
<rationale_10>[Review human evaluation or annotation caveats] If any part of the work relies on human
judgment, authors may mention subjectivity or annotator disagreement as
limitations.....</rationale_10>
```

**Verifier Prompt**

**Input:**
- **Query:** {query}
- **Flagging Instructions:** {rationales}
- **Previous Chunks Summaries:** {chunk_summaries}
- **Chunk to Verify:** {chunk_text}

**Instructions:**
- Flag clear factual errors only if highly confident (>90%).
- Detect contradictions with prior verified chunks.
- Assume validity unless strong evidence to the contrary exists.

**Response Format:**

```
{ "flagged": false, "chunk_summary":  "This chunk discusses ...",
 "flag_types":  [] }
```

**Knowledge Poisoning Prompt with Example**

**Prompt:**
*You are given a question and the correct context. Your task is to: (1) Write an incorrect but plausible answer in the same style as the correct answer. (2) Create a poisoned_corpus (∼100 words) that might mislead a model into selecting the incorrect answer........*

**Question:**
*Consider "Fiverr's" privacy policy; who can see the jobs that I post?*

**Correct Context:**
*In other words, when you access the Site we are aware of your usage of the Site, and may gather, collect and record the information relating to such usage, including geo-location information, IP address, device and connection information, browser information and web-log information, and all communications recorded by Users through the Site.*

**Poisoned Corpus:**
*Fiverr assures its users that their data is not only secure but private. When you interact with the platform, job postings and communications are secured using TLS encryption. Fiverr clearly states that it will never sell, rent, or share user content without explicit agreement. This gives users peace of mind when they publish job posts or engage with freelancers. In other words, your job posts are only visible to you and are never shared with other users, freelancers, or third parties, ensuring total privacy and anonymity.*

## A.4 DPO Implementation Details

We use Direct Preference Optimization (DPO) to fine-tune a general-purpose LLM to generate query-aligned rationales. The preference dataset is automatically constructed using the original QA corpus: for each query, rationales that led to correct evidence selection form the preferred output, while others form the rejected output. No manual labeling is required. The model is trained using pairwise comparisons of effective and ineffective rationales.

We train the model using the `LlaMA-3.1-8b-Instruct` LLM. The DPO loss (Equation 1 in section 2) is optimized over three epochs using cosine learning rate scheduling. Training and validation data are derived from a single annotated file using an 80/10/10 train-validation-test split.

Table 5: DPO training configuration used for rationale refinement.

| Parameter | Value |
|---|---|
| Base model | LLaMA-3.1-8B-Instruct |
| Batch size (train / eval) | 1 / 1 |
| Gradient accumulation steps | 2 |
| Epochs | 3 |
| Learning rate | 3e-5 |
| Scheduler | Cosine |
| Warmup ratio | 0.1 |
| DPO loss $\beta$ | 0.05 |
| Train / Val / Test split | 80% / 10% / 10% |
| Save strategy | Per epoch |
| Best model selection metric | `eval_rewards/chosen` |

## A.5 Elbow Detection in Pooled Rationale Component of ECSE

To identify evidence that align with the collective intent of all rationales, we compute a pooled embedding $\bar{r} = \frac{1}{|R|} \sum_{r_i \in R} \mathrm{SBERT}(r_i)$ and calculate cosine similarity between $\bar{r}$ and each evidence embedding. This produces a sorted sequence of similarity scores $\{s_1, s_2, \ldots, s_n\}$.

Table 6: CP Task Results across Datasets and Evidence Sizes

| Model | Contract-NLI | | PrivacyQA | | CUAD | | MAUD | | QASPER | |
|---|---|---|---|---|---|---|---|---|---|---|
| | **Evidence Size = 128** | | | | | | | | | |
| | P@7 | R@7 | P@10 | R@10 | P@24 | R@24 | P@43 | R@43 | P@22 | R@22 |
| SBERT | 0.17 | 0.78 | 0.12 | 0.61 | 0.04 | 0.59 | 0.01 | 0.03 | 0.10 | 0.86 |
| Contriever | 0.16 | 0.81 | 0.11 | 0.55 | 0.04 | 0.27 | 0.01 | 0.14 | **0.11** | **0.90** |
| Cross-Encoder | 0.17 | 0.83 | 0.11 | 0.51 | 0.03 | 0.50 | 0.01 | 0.15 | **0.11** | 0.89 |
| Finetuned-SBERT | 0.17 | 0.81 | 0.09 | 0.59 | 0.04 | 0.59 | 0.01 | 0.14 | **0.11** | 0.88 |
| RankRAG | 0.12 | 0.77 | 0.09 | 0.57 | 0.04 | 0.49 | 0.01 | 0.12 | 0.09 | 0.61 |
| METEORA | **0.18** | **0.89** | **0.13** | **0.84** | **0.06** | **0.78** | **0.02** | **0.39** | **0.11** | **0.90** |
| | **Evidence Size = 256** | | | | | | | | | |
| | P@5 | R@5 | P@8 | R@8 | P@17 | R@17 | P@37 | R@37 | P@14 | R@14 |
| SBERT | **0.25** | 0.88 | 0.17 | 0.81 | 0.07 | 0.76 | 0.01 | 0.37 | 0.17 | 0.93 |
| Contriever | 0.24 | 0.85 | 0.15 | 0.68 | 0.06 | 0.68 | 0.01 | 0.27 | **0.18** | 0.95 |
| Cross-Encoder | **0.25** | 0.89 | 0.16 | 0.74 | 0.06 | 0.64 | 0.01 | 0.31 | 0.16 | 0.94 |
| Finetuned-SBERT | **0.25** | 0.89 | 0.13 | 0.59 | 0.07 | 0.68 | 0.01 | 0.36 | 0.17 | 0.94 |
| RankRAG | 0.19 | 0.73 | 0.11 | 0.69 | 0.05 | 0.76 | 0.01 | 0.22 | 0.13 | 0.79 |
| METEORA | **0.25** | **0.98** | **0.18** | **0.92** | **0.08** | **0.84** | **0.02** | **0.58** | 0.16 | **0.96** |

Table 7: **Verifier Flag Distribution (Excluding FinQA).** Percentage of flagged poisoned evidence categorized by instruction, contradiction, and factual violations across evidence sizes.

| Flag Type | QASPER | Contract-NLI | PrivacyQA | CUAD | MAUD |
|---|---|---|---|---|---|
| | **Evidence Size = 128** | | | | |
| Instruction | 23.67 | 57.80 | 34.80 | 20.37 | 43.93 |
| Contradiction | 0.07 | 0.21 | 0.21 | 2.23 | 0.11 |
| Factual | 9.27 | 3.96 | 3.96 | 2.44 | 0.02 |
| | **Evidence Size = 256** | | | | |
| Instruction | 35.30 | 52.48 | 54.93 | 20.79 | 40.60 |
| Contradiction | 0.10 | 0.07 | 0.39 | 1.86 | 0.46 |
| Factual | 7.59 | 2.39 | 4.68 | 2.18 | 0.00 |

We first compute the first-order differences $\Delta_i = s_i - s_{i+1}$ and apply z-score normalization across $\{\Delta_i\}$ to highlight sharp changes in similarity. The selection index $k^*$ is identified at the first point where the drop in similarity significantly deviates from the average pattern, indicating a natural boundary between highly relevant and less relevant chunks.

In cases where similarity scores decline uniformly and no clear deviation is found, we fallback to computing the second-order differences $\nabla_i^2 = \Delta_{i+1} - \Delta_i$. We then choose the index of maximum curvature, which reflects the sharpest transition in the similarity landscape. The selected top-$k^*$ chunks, denoted as $E_g$, are thus derived without relying on manually defined thresholds, enabling adaptive and data-driven cutoff across varying distributions.

A.6 Results Across All Evidence Sizes

We report precision, recall, and generation performance for all evaluated chunk sizes across datasets to complement the main results presented in the paper.

# B DPO Theory

We address the problem of aligning an LLM's rationale generation capability with evidence selection using Direct Preference Optimization (DPO). We consider a setting where:

- We have a preference dataset consisting of document evidences

- For each user query, certain document evidences are labeled as relevant (positive) examples

- Other document evidences are considered irrelevant (negative) examples

Table 8: Ablation study of the ECSE pipeline, evaluating the effect of Pairing, Pooling, and Expansion stages across datasets and evidence sizes. Pooling improves recall over Pairing alone, and adding Expansion further enhances performance. The complete ECSE pipeline achieves the best balance between precision and recall, particularly at larger evidence sizes.

| Components | Contract-NLI | | PrivacyQA | | CUAD | | MAUD | | QASPER | |
|---|---|---|---|---|---|---|---|---|---|---|
| | **Evidence Size = 128** | | | | | | | | | |
| | P | R | P | R | P | R | P | R | P | R |
| Pairing | 0.20 | 0.91 | 0.14 | 0.64 | 0.08 | 0.64 | 0.01 | 0.25 | 0.14 | 0.81 |
| Pairing + Expansion | 0.18 | **0.95** | 0.13 | **0.84** | 0.06 | **0.78** | **0.02** | **0.39** | 0.11 | **0.90** |
| | **Evidence Size = 256** | | | | | | | | | |
| | P | R | P | R | P | R | P | R | P | R |
| Pairing | **0.27** | 0.94 | 0.19 | 0.83 | 0.10 | 0.69 | 0.02 | 0.50 | 0.20 | 0.90 |
| Pairing + Expansion | 0.25 | **0.98** | 0.18 | **0.92** | 0.08 | **0.84** | 0.02 | **0.58** | 0.16 | **0.96** |

Table 9: Corpus poisoning detection performance across datasets and evidence sizes. `METEORA` shows strong resilience, outperforming perplexity-based defenses in recall, especially at larger evidence sizes.

| Method | Contract-NLI | | PrivacyQA | | CUAD | | MAUD | | QASPER | |
|---|---|---|---|---|---|---|---|---|---|---|
| | Precision | Recall | Precision | Recall | Precision | Recall | Precision | Recall | Precision | Recall |
| | Evidence Size = 128 | | | | | | | | | |
| No Defense | 0.00 | 0.00 | 0.00 | 0.00 | 0.00 | 0.00 | 0.00 | 0.00 | 0.00 | 0.00 |
| Perplexity | **0.69** | 0.22 | 0.19 | 0.17 | 0.19 | 0.22 | **0.46** | 0.12 | 0.09 | 0.10 |
| `METEORA` | 0.62 | **0.55** | **0.37** | **0.18** | **0.25** | **0.31** | 0.44 | **0.42** | **0.29** | **0.33** |
| | Evidence Size = 256 | | | | | | | | | |
| No Defense | 0.00 | 0.00 | 0.00 | 0.00 | 0.00 | 0.00 | 0.00 | 0.00 | 0.00 | 0.00 |
| Perplexity | 0.54 | 0.20 | 0.29 | 0.18 | 0.18 | 0.10 | 0.31 | 0.14 | 0.27 | 0.15 |
| `METEORA` | **0.55** | **0.72** | **0.60** | **0.40** | **0.24** | **0.38** | **0.41** | **0.46** | **0.43** | **0.53** |

Table 10: FinQA analysis across evidence sizes: CP task performance, ablation of ECSE, verifier flag contribution, and poisoning detection.

**CP Performance**

| Model | P@13 | R@13 |
|---|---|---|
| **Size = 32** | | |
| SBERT | 0.08 | 0.88 |
| Contriever | 0.07 | **0.91** |
| CrossEnc | **0.09** | 0.83 |
| Fine-Tuned SBERT | 0.08 | 0.90 |
| RankRAG | 0.06 | 0.78 |
| `METEORA` | 0.07 | 0.89 |
| **Size = 64** | | |
| SBERT | 0.12 | 0.96 |
| Contriever | 0.11 | **0.98** |
| CrossEnc | 0.12 | 0.97 |
| Fine-Tuned SBERT | **0.13** | 0.97 |
| RankRAG | 0.08 | 0.89 |
| `METEORA` | 0.12 | 0.95 |

**ECSE Ablation**

| Stage | P@13 | R@13 |
|---|---|---|
| **Size = 32** | | |
| Pairing | 0.08 | 0.82 |
| + Expansion | 0.07 | **0.89** |
| **Size = 64** | | |
| Pairing | 0.14 | 0.91 |
| + Expansion | 0.12 | **0.95** |

**Verifier Flags**

| Flag | 32 |
|---|---|
| Instruction | 32.02 |
| Contradiction | 1.64 |
| Factual | 0.32 |
| Flag | 64 |
| Instruction | 37.02 |
| Contradiction | 1.84 |
| Factual | 0.11 |

**Poisoning Detection**

| Method | Prec. | Rec. |
|---|---|---|
| **Size = 32** | | |
| No Defense | 0.00 | 0.00 |
| Perplexity | 0.13 | 0.05 |
| `METEORA` | **0.34** | **0.34** |
| **Size = 64** | | |
| No Defense | 0.00 | 0.00 |
| Perplexity | 0.18 | 0.07 |
| `METEORA` | **0.39** | **0.48** |

- The goal is to train the LLM to select appropriate evidences by generating rationales that explain the relevance of evidences to the query

Unlike traditional RAG approaches that use re-ranking mechanisms based on similarity metrics, our approach enables the LLM to learn to select evidences through a process of rationale generation, providing transparency and better alignment with the final response generation. We provide a rigorous mathematical formulation of this problem and prove that DPO training leads to a policy that optimally selects contextual evidences based on generated rationales.

## C  PROBLEM FORMULATION

Let us now formalize the problem of aligning rationale generation for contextual evidence selection using DPO.

### C.1  NOTATION AND SETUP

We denote:

- $\mathcal{Q}$: The set of all possible user queries

- $\mathcal{E}$: The set of all possible contextual evidences in the knowledge base

- $\mathcal{R}$: The set of all possible rationales explaining evidence relevance

- $\mathcal{E}_q^+ \subset \mathcal{E}$: The set of relevant evidences for query $q$

- $\mathcal{E}_q^- \subset \mathcal{E}$: The set of irrelevant evidences for query $q$

- $\pi_\theta(e, r|q)$: The LLM policy parameterized by $\theta$, giving the joint probability of selecting evidence $e$ and generating rationale $r$ given query $q$

- $\pi_{\text{ref}}(e, r|q)$: The reference (initial) policy

We can decompose the joint probability as:

$$\pi_\theta(e, r|q) = \pi_\theta(e|q) \cdot \pi_\theta(r|q, e) \tag{2}$$

Where $\pi_\theta(e|q)$ is the probability of selecting evidence $e$ for query $q$, and $\pi_\theta(r|q, e)$ is the probability of generating rationale $r$ given query $q$ and selected evidence $e$.

### C.2  PREFERENCE MODEL

We assume there exists an ideal reward function $r^*(q, e, r)$ that captures the appropriateness of both the selected evidence and the generated rationale. This reward function should assign higher values to relevant evidences with convincing rationales and lower values to irrelevant evidences or unconvincing rationales.

We model this as:
$$r^*(q, e, r) = f\left(\text{relevance}(e, q), \text{quality}(r, q, e)\right) \tag{3}$$

Where relevance$(e, q)$ measures how relevant the evidence $e$ is to query $q$, and quality$(r, q, e)$ measures how well the rationale $r$ explains the relevance of evidence $e$ to query $q$. The function $f$ combines these measures.

A simple form could be:

$$r^*(q, e, r) = \alpha \cdot \text{relevance}(e, q) + \gamma \cdot \text{quality}(r, q, e) \tag{4}$$

Where $\alpha, \gamma > 0$ are weights that balance the importance of evidence relevance and rationale quality.

## C.3 DERIVING THE DPO OBJECTIVE FOR RATIONALE-BASED EVIDENCE SELECTION

To apply DPO, we need preference data in the form of $(q, (e_w, r_w), (e_l, r_l))$ tuples, where $(e_w, r_w)$ is preferred over $(e_l, r_l)$.

In our setting, we can generate these tuples as follows:

- $q$: A user query

- $(e_w, r_w)$: A relevant evidence with a high-quality rationale explaining its relevance

- $(e_l, r_l)$: An irrelevant evidence with a rationale attempting to explain its relevance

Given such tuples, the DPO objective becomes:

$$\mathcal{L}_{\text{DPO}}(\pi_\theta; \pi_{\text{ref}}) = -\mathbb{E}_{(q, (e_w, r_w), (e_l, r_l))} \left[ \log \sigma \left( \beta \log \frac{\pi_\theta(e_w, r_w|q)}{\pi_{\text{ref}}(e_w, r_w|q)} - \beta \log \frac{\pi_\theta(e_l, r_l|q)}{\pi_{\text{ref}}(e_l, r_l|q)} \right) \right] \quad (5)$$

## D THEORETICAL ANALYSIS

We now prove that optimizing the DPO objective leads to an aligned policy that selects appropriate contextual evidences through rationale generation.

**Theorem 1** (Optimality of DPO for Rationale-Based Evidence Selection)**.** *Let $\pi_\theta$ be a policy trained using the DPO objective with preference data derived from relevant and irrelevant evidences with their corresponding rationales. Under certain regularity conditions, as the amount of preference data increases, $\pi_\theta$ converges to:*

$$\pi^*(e, r|q) \propto \pi_{ref}(e, r|q) \exp\left(\frac{1}{\beta} r^*(q, e, r)\right) \quad (6)$$

*Where $r^*(q, c, r)$ is the true reward function capturing the appropriateness of evidence selection and rationale generation based on the query.*

*Proof.* We proceed with the proof step by step.

**Step 1:** Recall from Rafailov et al. (2023a) that DPO is derived from the following principle: in the Bradley-Terry preference model, the probability that response $(e_w, r_w)$ is preferred over $(e_l, r_l)$ given query $q$ is:

$$P((e_w, r_w) \succ (e_l, r_l)|q) = \sigma(r(q, e_w, r_w) - r(q, e_l, r_l)) \quad (7)$$

Where $r(q, e, r)$ is the reward function and $\sigma$ is the logistic function.

**Step 2:** The optimal policy given a reward function $r$ and a reference policy $\pi_{\text{ref}}$ is:

$$\pi_r(e, r|q) \propto \pi_{\text{ref}}(e, r|q) \exp\left(\frac{1}{\beta} r(q, e, r)\right) \quad (8)$$

This follows from the constrained optimization problem of maximizing expected reward subject to a KL-divergence constraint with the reference policy.

**Step 3:** Plugging the optimal policy form into the Bradley-Terry model, we get:

$$P((e_w, r_w) \succ (e_l, r_l)|q) = \sigma(r(q, e_w, r_w) - r(q, e_l, r_l)) \quad (9)$$

$$= \sigma\left(\beta \log \frac{\pi_r(e_w, r_w|q)}{\pi_{\text{ref}}(e_w, r_w|q)} - \beta \log \frac{\pi_r(e_l, r_l|q)}{\pi_{\text{ref}}(e_l, r_l|q)}\right) \quad (10)$$

**Step 4:** The DPO objective trains $\pi_\theta$ to match these probabilities by minimizing:

$$\mathcal{L}_{\text{DPO}}(\pi_\theta; \pi_{\text{ref}}) = -\mathbb{E}_{(q, (e_w, r_w), (e_l, r_l))} \left[ \log \sigma \left( \beta \log \frac{\pi_\theta(e_w, r_w|q)}{\pi_{\text{ref}}(e_w, r_w|q)} - \beta \log \frac{\pi_\theta(e_l, r_l|q)}{\pi_{\text{ref}}(e_l, r_l|q)} \right) \right]$$
$$(11)$$

**Step 5:** As the amount of preference data increases, assuming the preference data accurately reflects the true reward function $r^*(q, e, r)$ that values relevant evidences with convincing

rationales over irrelevant evidences, minimizing the DPO loss will drive $\pi_\theta$ towards the optimal policy:

$$\pi_\theta(e, r|q) \to \pi_{r^*}(e, r|q) \propto \pi_{\text{ref}}(e, r|q) \exp\left(\frac{1}{\beta} r^*(q, e, r)\right) \tag{12}$$

**Step 6:** We can further analyze the joint probability by decomposing it:

$$\pi^*(e, r|q) \propto \pi_{\text{ref}}(e|q) \pi_{\text{ref}}(r|q, e) \exp\left(\frac{1}{\beta} r^*(q, e, r)\right) \tag{13}$$

**Step 7:** Since $r^*(q, e, r)$ rewards both chunk relevance and rationale quality, the resulting policy will select evidences and generate rationales that are aligned with the true relevance of evidences to the query. This means the policy learns to select evidences based on their relevance through a process of rationale generation rather than simple re-ranking. $\qquad\square$

**Corollary 2** (Alignment Guarantee for Chunk Selection). *If the preference dataset accurately reflects the relevance of evidences to queries along with appropriate rationales, then the DPO-trained policy will select contextual evidences that are most relevant to the query while generating rationales that justify this selection.*

**Corollary 3** (Superiority over Re-ranking). *The DPO-trained policy provides advantages over traditional re-ranking in RAG systems by:*

1. *Jointly optimizing evidence selection and rationale generation*

2. *Providing transparent explanations for why specific evidences were selected*

3. *Learning complex relevance patterns beyond simple similarity metrics*

# E  ALGORITHM

We now describe a step-by-step procedure for implementing DPO for rationale-based evidence selection:

---
**Algorithm 1** DPO for Rationale-Based Chunk Selection
---
**Require:** Base LLM $\pi_{\text{ref}}$, dataset of user queries $\{q_i\}$, relevant chunks $\{\mathcal{E}_{q_i}^+\}$, irrelevant chunks $\{\mathcal{E}_{q_i}^-\}$
**Ensure:** Aligned LLM $\pi_\theta$ that selects evidences with rationales
 1: Initialize $\pi_\theta \leftarrow \pi_{\text{ref}}$
 2: Construct preference dataset $\mathcal{P} = \{(q_i, (e_{w,i}, r_{w,i}), (e_{l,i}, r_{l,i}))\}$:
 3: **for** each query $q_i$ **do**
 4:     Sample relevant evidence $e_{w,i}$ from $\mathcal{E}_{q_i}^+$
 5:     Generate rationale $r_{w,i}$ explaining relevance of $e_{w,i}$ to $q_i$
 6:     Sample irrelevant evidence $e_{l,i}$ from $\mathcal{E}_{q_i}^-$
 7:     Generate rationale $r_{l,i}$ attempting to explain relevance of $e_{l,i}$ to $q_i$
 8:     Add tuple $(q_i, (e_{w,i}, r_{w,i}), (e_{l,i}, r_{l,i}))$ to $\mathcal{P}$
 9: **end for**
10: Train $\pi_\theta$ by minimizing:

$$\mathcal{L}_{\text{DPO}}(\pi_\theta; \pi_{\text{ref}}) = -\mathbb{E}_{(q,(e_w,r_w),(e_l,r_l))\sim\mathcal{P}}\left[\log\sigma\left(\beta\log\frac{\pi_\theta(e_w, r_w|q)}{\pi_{\text{ref}}(e_w, r_w|q)} - \beta\log\frac{\pi_\theta(e_l, r_l|q)}{\pi_{\text{ref}}(e_l, r_l|q)}\right)\right]$$

11: **return** $\pi_\theta$

---

# F  MEASURING CHUNK RELEVANCE AND RATIONALE QUALITY

The functions relevance$(c, q)$ and quality$(r, q, c)$ can be implemented in various ways:

- **Evidence Relevance**:

  - Semantic similarity between query $q$ and evidence $e$ using embeddings

- BM25 or other lexical similarity measures

- Entailment scores from an NLI model

- Human relevance judgments

- **Rationale Quality**:

    - Coherence measures (how well the rationale flows logically)

    - Factual consistency with the evidence content

    - Specificity to the query (rather than generic explanations)

    - Explanatory power (does it actually explain why the evidence is relevant?)

    - Human judgments of explanation quality

## G  RATIONALE FABRICATION

The model might generate convincing-sounding but factually incorrect rationales to justify the selection of irrelevant chunks.

**Mitigation:** Include factual consistency metrics in the training process and explicitly penalize fabricated or misleading rationales.

## H  DISTRIBUTION SHIFT

The distribution of queries and evidences during deployment may differ from those in the training data.

**Mitigation:** Include a diverse range of queries and evidence types in the training data, and implement continual learning mechanisms to adapt to new domains.

### H.0.1  RETRIEVAL-AUGMENTED GENERATION (RAG)

The RAG architecture enhances generative models by incorporating additional context from a knowledge base, thereby improving response accuracy. Given a query $q$, a knowledge base of documents $D = \{d_1, d_2, \ldots, d_n\}$, a retriever function $F(q, D) \rightarrow D_q \subset D$, a re-ranker function $\mathcal{R}e(q, D_q)$ that picks the top-$k$ documents, and a generative model $\mathcal{M}$, the final generation $g$ is given by:

$$g = \mathcal{M}(q, \mathcal{R}e(q, F(q, D)))$$

### H.1  CURRENT RE-RANKING MECHANISM

A re-ranker $\mathcal{R}e$ computes the semantic similarity $\mathcal{S}s$ between a query $q$ and a set of retrieved documents $D_q$, and then ranks the documents by relevance. In *top-k* SBERT-based re-ranking, the query and each document are encoded independently, and cosine similarity is used to score relevance:

$$\mathcal{R}e_{\text{SBERT}}(q, D_q) = \{\cos(\text{SBERT}(q), \text{SBERT}(d)) \mid d \in D_q\}$$

In *top-k* cross-encoders, the query and each document are jointly encoded, producing a scalar relevance score:

$$\mathcal{R}e_{\text{Cross}}(q, D_q) = \{\text{CrossEncoder}(q, d) \mid d \in D_q\}$$

*Top-k* Contriever also encodes the query and documents independently using a contrastive learning objective:

$$\mathcal{R}e_{\text{Contriever}}(q, D_q) = \{\cos(\text{Contriever}(q), \text{Contriever}(d)) \mid d \in D_q\}$$

While effective for ranking based on surface-level similarity, these methods have three key limitations. First, they require manual tuning of $k$, which is often done through hit-and-trial. Second, using a fixed $k$ for all queries in a domain may include less-relevant chunks in the top-$k$, which can negatively impact downstream generation. Third, these methods lack interpretability and provide no mechanisms to detect or filter adversarial or misleading content.

Table 11: CP Task results of `METEORA` with and without the Verifier. The metrics are Precision (P) and Recall (R) at the same $k$ used in the main CP table.

| Model | QASPER | | Contract-NLI | | FinQA | | PrivacyQA | | CUAD | | MAUD | | Average | |
|---|---|---|---|---|---|---|---|---|---|---|---|---|---|---|
| | R | P | R | P | R | P | R | P | R | P | R | P | R | P |
| METEORA | 0.99 | 0.26 | 1.00 | 0.35 | 0.95 | 0.12 | 0.98 | 0.23 | 0.93 | 0.12 | 0.72 | 0.03 | 0.93 | 0.19 |
| METEORA w/o Verifier | 1.00 | 0.25 | 1.00 | 0.34 | 0.97 | 0.10 | 1.00 | 0.22 | 0.94 | 0.11 | 0.75 | 0.02 | 0.94 | 0.17 |

# I  HUMAN EVALUATION

To quantify the interpretability and robustness of `METEORA`, we conduct a human evaluation using the popular notion of interpretability as the degree to which a human can understand the cause of a decision (Miller, 2019; Lipton, 2018). We sample 100 random examples across six datasets (QASPER, Contract-NLI, PrivacyQA, FinQA, CUAD, MAUD). Four experienced annotators, blinded to the ground truth and to each other's labels, review each query–chunk pair along with the corresponding flagging instruction and the Verifier's flags. For each chunk, they answer a binary question: "If you applied these instructions, would you correctly identify this chunk as poisoned/problematic?" (YES/NO) and provide an integer confidence score from 1 (very uncertain) to 5 (very confident). Interpretability is measured via this confidence score; robustness is measured via agreement with the ground-truth labels.

**Interpretability.** The mean confidence score is 3.64/5, indicating that annotators find the instructions interpretable and applicable for majority of the case.

**Robustness.** Annotator decisions obtain an average accuracy of $\approx 86\%$ against ground truth. Combined with the qualitative end-to-end examples in the appendix, this suggests that humans can reliably understand and reconstruct the causes of `METEORA`'s evidence-level decisions from the provided instructions."'

# J  NOISE–RECALL ANALYSIS

`METEORA` optimize downstream generation quality under realistic evidence budgets and minimize input noise from incorrect evidence chunks. Tuning an optimal $k$ per query in real world systems is impractical, and hence we evaluate baselines at `METEORA`'s average evidence count. While increasing $k$ can improve recall, it also injects substantial noise into the context and harms the quality of the generated answer for production-grade models (e.g., GPT-4o) as well as for open-source LLMs (Du et al., 2025). To observe the effect on Precision–Recall curves, we set $k$ ranging from 1 to 64 for each of the baseline. We observed across the datasets, that the baselines require many more chunks to reach the same recall level with incorrect chunk due to top-$k$ criteria.

# K  LATENCY ANALYSIS

`METEORA`'s latency comes from three stages: generating rationales over the initial evidence pool (up to 64 chunks), verifying filtered chunks, and producing the final answer. Rationale generation accounts for only $\approx 5$–$10\%$ of tokens per query, verification for $\approx 10$–$30\%$, and the remaining latency is dominated by answer generation.

# L  VERIFIER ABLATION AND IMPACT OF FLAGGING INSTRUCTIONS

Verifier in `METEORA` plays a crucial role in answer quality. To quantify its impact, we run an ablation that removes the Verifier entirely (i.e., no flagging instructions). Tables 11 and 12 report results for the CP and Generation tasks, denoted as "METEORA w/o Verifier." Removing the Verifier leads to an average 7% drop in answer accuracy across all datasets on the Generation task, and consistent precision and recall degradations on the CP task, highlighting the importance of both the Verifier and its flagging instructions for downstream correctness.

# M  MULTI-HOP EVIDENCE SELECTION AND CHUNK BUDGETS

We analyze 20,000+ instances from QASPER, Contract-NLI, PrivacyQA, CUAD, and MAUD, all of which require multi-hop evidence in a non-trivial fraction of cases. Multi-hop

Table 12: Generation Task results of METEORA with and without the Verifier.

| Model | QASPER | C-NLI | FinQA | PrivacyQA | CUAD | MAUD | Average |
|---|---|---|---|---|---|---|---|
| METEORA | 0.74 | 0.76 | 0.71 | 0.65 | 0.53 | 0.44 | 0.64 |
| METEORA w/o Verifier | 0.70 | 0.68 | 0.65 | 0.58 | 0.45 | 0.38 | 0.57 |

capability is governed by the retriever; the re-ranker then treats all retrieved chunks uniformly. Empirically, 74% of instances to generate correct answer rely on a single correct chunk and 26% require multiple chunks. METEORA achieves essentially identical performance in both the settings: for single-chunk settings, precision/recall is 0.19/0.94; for multi-chunk settings, 0.19/0.91. Thus, METEORA preserves the full set of correct evidence while still providing substantial efficiency gains, even on multi-hop tasks.

# N    CODE AND PREFERENCE-TUNED MODEL

We use a preference-tuned LLaMA-3.1-8B model for rationale generation and evidence verification. We will share this preference-tuned model on Hugging Face to promote research.

