# OpenReview forum: "Ranking Free RAG: Replacing Re-ranking with Selection in RAG for Sensitive Domains"
_ICLR.cc/2026/Conference — ICLR 2026 Conference Desk Rejected Submission_

### Official Review · Reviewer_E7Tq · 2025-10-23

**Soundness:** 2
**Presentation:** 3
**Contribution:** 1
**Rating:** 4
**Confidence:** 3

**Summary:**

The paper introduces METEORA, a ranking-free RAG pipeline that (i) DPO-tunes a rationale generator, (ii) uses a two-step Evidence Chunk Selection Engine with elbow detection and optional neighborhood expansion, and (iii) reuses the same rationales in a Verifier LLM to filter poisoned/irrelevant evidence before answer generation.

**Strengths:**

1. Replaces opaque top-k heuristics with rationale-driven selection; the same rationale frame powers selection and verification, improving auditability for sensitive domains.

2. Easy to implement and tune.

3. Breadth of evaluation, three tasks across six long-document datasets

**Weaknesses:**

1. In CP, baselines are evaluated at METEORA’s average evidence count rather than each method’s own best-K; a full K-sweep with per-method optima would be more standard and may change relative standings.

2. Design-wise, selection and verification share the same rationale frame, which authors themselves tag as a single-point-of-failure risk under targeted attacks; some verifier/model heterogeneity would help.

3. FinQA case shows lower recall than re-rankers in short-passage settings; the paper attributes this to construction choices, but a deeper failure analysis (tables, chunking, rationale style) would clarify when selection loses to re-ranking.

**Questions:**

1. Section 2.3 (Verifier): Do verifier flags correlate with downstream answer correctness on each dataset? Please report correlations/AUCs, not just counts.

2. What are your failure modes when elbow detection under- or over-selects? Any guardrails you recommend (min/max caps, second-pass knee tests)?

3. For typical r and pool sizes, what fraction of tokens/latency goes to rationale generation and to verifier passes?

---

> ### Author Response · Authors · 2025-11-21
> **Authors' Response 1**
>
> We thank the reviewer E7Tq for acknowledging the breadth of evaluation.
>
> > **W1**. In CP, baselines are evaluated at METEORA’s average evidence count rather than each method’s own best-K; a full K-sweep with per-method optima would be more standard and may change relative standings.
>
> **A1** METEORA helps to improve downstream generation quality by minimizing input noise. We evaluate baselines at METEORA's average evidence count, as finding an optimal k for every specific query is impractical in real-world systems. While increasing k artificially boosts recall, it introduces significant noise, directly undermining the system's core objective: accurate generation as described by Du et al. [1]. Such a performance drop is seen in both production-grade models, e.g., GPT-4o, as well as open-source models. Therefore, evaluating at an artificially high "best-k" penalizes the actual goal of the RAG pipeline.
>
> For a full comparison, we have included Precision-Recall (P-R) curves (Figure [link](https://anonymous.4open.science/r/METEORA-DC46/README.md)) with k from 1 to 64. These figures explicitly mark the k required for baselines to match METEORA's recall, demonstrating that METEORA achieves comparable coverage with 80% less noise.
>
> **References:**
>
> [1] Yufeng Du, Minyang Tian, Srikanth Ronanki, Subendhu Rongali, Sravan Babu Bodapati, Aram Galstyan, Azton Wells, Roy Schwartz, Eliu A Huerta, and Hao Peng. 2025. Context Length Alone Hurts LLM Performance Despite Perfect Retrieval. In Findings of the Association for Computational Linguistics: EMNLP 2025, pages 23281–23298, Suzhou, China. Association for Computational Linguistics.
>
> > **W2**: Design-wise, selection and verification share the same rationale frame, which authors themselves tag as a single-point-of-failure risk under targeted attacks; some verifier/model heterogeneity would help.
>
> **A2**. METEORA is flexible, and it is not restrictive to use the same rational frame for selection and verification. METEORA allows for any LLM model (and combinations of different model families) to be used for the Rationale Generator, the Verifier LLM, and the Answer Generator. This flexibility allows users to easily introduce heterogeneity to address the single-point-of-failure risk. Moreover, METEORA is fully compatible with any defense strategy for adversarial attacks studied in prior research. METEORA's framework for evidence selection can be augmented by existing adversarial defense techniques applied to the model components.
>
> We have explicitly identified this as a limitation in the footnote of Page 6 of the paper and will make it explicit in the final paper.
>
> > **W3**. FinQA case shows lower recall than re-rankers in short-passage settings; the paper attributes this to construction choices, but a deeper failure analysis (tables, chunking, rationale style) would clarify when selection loses to re-ranking.
>
> **A3**: FinQA contains correct answers directly in a short document, and traditional re-rankers (which rely on simple query-passage similarity) are slightly better and more effective than METEORA. METEORA is more effective at maintaining high recall while significantly reducing noise in a long-document setting.
>
> > **Q1**: Section 2.3 (Verifier): Do verifier flags correlate with downstream answer correctness on each dataset? Please report correlations/AUCs, not just counts.
>
> **A1**: Yes, flagging instructions affect evidence selection. We ran a new ablation without the Verifier (that is, without flagging instructions), which resulted in a **7%** drop in accuracy, as shown in Table R2. This highlights the correlation between verifier flags and downstream answer correctness.
>
> | Model                | QASPER             | C-NLI              | FinQA              | PrivacyQA          | CUAD               | MAUD               | Average             |
> |---|----|---|----|----|-------|-------|-----|
> | METEORA              | P: 0.26, R: 0.99   | P: 0.35, R: 1.00   | P: 0.12, R: 0.95   | P: 0.23, R: 0.98   | P: 0.12, R: 0.93   | P: 0.03, R: 0.72   | P: **0.19**, R: 0.93    |
> | METEORA w/o Verifier | P: 0.25, R: 1.00   | P: 0.34, R: 1.00   | P: 0.10, R: 0.97   | P: 0.22, R: 1.00   | P: 0.11, R: 0.94   | P: 0.02, R: 0.75   | P: 0.17, R: **0.94**    |
>
> **Table R1:** CP Task results of METEORA without Verifier. Here P = Precision, R = Recall.
>
> ----
>
> | Model                | QASPER | C-NLI | FinQA | PrivacyQA | CUAD | MAUD | Average |
> |-------|:------:|:-----:|:-----:|:------:|:----:|:----:|:-----:|
> | METEORA              |  0.74  | 0.76  | 0.71  |   0.65    | 0.53 | 0.44 |  **0.64**   |
> | METEORA w/o Verifier |  0.70  | 0.68  | 0.65  |   0.58    | 0.45 | 0.38 |  0.57   |
>
> **Table R2:** Generation Task results of METEORA without Verifier

---

> > ### Author Response · Authors · 2025-11-21
> > **Authors' Response 2**
> >
> > > **Q2** What are your failure modes when elbow detection under- or over-selects? Any guardrails you recommend (min/max caps, second-pass knee tests)?
> >
> > **A2**: We partially address failure modes in the paper through both the design of ECSE and the fallback behaviors built into the cutoff mechanism. In §2.2 (lines 246–269), we detail the z-score–based elbow detector and explicitly describe how cutoff determination depends on the geometry of the similarity curve. We also provide a fallback when no strong elbow exists: if all z-scores are below the threshold, we switch to a second-order curvature test (lines 262–265) to identify the point of maximum slope change. This is the first safeguard against pathological under- or over-selection. Further, the motivation for adaptive selection “without fixed top-k constraints” (lines 189–195) highlights the need for robustness across variable evidence-curve shapes.
> >
> > We recommend interpretable guardrails consistent with our ranking-free philosophy:
> >
> > 1. Adaptive minimum/maximum caps on the number of selected chunks (e.g., ensuring at least one chunk per rationale, or bounding expansion to avoid runaway selection).
> > 2. A second-pass knee or curvature test is a second pass that re-examines the curve’s shape when the initial elbow test does not identify a clear cutoff (lines 262–265).
> > 3. Light neighbor expansion (lines 266–269) to prevent under-selection in multi-chunk spans.
> >
> > > **Q3** For typical r and pool sizes, what fraction of tokens/latency goes to rationale generation and to verifier passes?
> >
> > **A3** Concretely, under the settings used in our experiments, evidence pools of n = 32–64, chunk size 512, and a small number of rationales r in the low single digits, the single rationale-generation call is a few dozen output tokens on top of the query prompt. That typically amounts to ≈5–10% of total tokens/latency per query, consistent with our statement that $L_{prefill}$ ≫ $L_{rat}$.
> >
> > The Verifier runs once per selected chunk e $\in$ $E_s$. However, $E_s$ is much smaller than the evidence sets used by baselines (often 3–5× fewer chunks), so verifier passes scale with this reduced *k*. In a typical configuration where you verify all selected chunks, verifier calls account for roughly 10–30% of tokens/latency, with the remaining 60%+ dominated by the final generation pass. In cleaner, non-adversarial deployments, it is possible to downsample or disable the Verifier, turning its fraction close to zero, while the rationale step remains a small, roughly single-digit percentage overhead on top of the main answer generation.

---

> ### Author Response · Authors · 2025-11-27
> **Follow-up to Reviewer E7Tq**
>
> Thank you for the thoughtful review and suggestions. We now include P-R curves over k, clearer discussion of FinQA and elbow/expansion failure modes, and a breakdown of rationale/Verifier overhead, and will surface these points in the main text. If this resolves your key concerns, we would appreciate it if you could consider updating your score.

---

### Official Review · Reviewer_RMUm · 2025-10-26

**Soundness:** 2
**Presentation:** 2
**Contribution:** 1
**Rating:** 2
**Confidence:** 3

**Summary:**

The paper aims to make RAG systems more auditable, interpretable, and robust in sensitive domains like law, finance, and healthcare. Instead of relying on opaque re-ranking and arbitrary top-k retrieval, it introduces METEORA, a framework that uses explicit rationales to guide evidence selection. A rationale generator trained with Direct Preference Optimization (DPO) produces interpretable reasoning sentences, which drive a two-stage Evidence Chunk Selection Engine to adaptively choose relevant evidence and optionally expand context. A Verifier LLM then filters out poisoned or misleading chunks. Experiments show that METEORA improves retrieval quality, downstream answer accuracy, and robustness against corpus poisoning.

**Strengths:**

It offers a fresh perspective on re-ranking by grounding it in explicit reasoning — using a rationale generator to make the evidence selection process more transparent and auditable, providing a cleaner and more interpretable alternative to standard top-k retrieval.

**Weaknesses:**

1. While METEORA shows clear performance gains, the paper doesn’t provide much concrete analysis or evidence to quantify its initial claims around improved interpretability and credibility. It would be helpful to see a more systematic evaluation of these aspects.

2. Methodologically, METEORA feels more like an engineering refinement built on existing techniques. The main novelty seems to be the unsupervised evidence selection approach, but its real advantages aren’t clearly demonstrated — the paper lacks ablation or comparative analysis to justify its effectiveness.

**Questions:**

1. Prior work has shown that correct answers don’t always come from correct reasoning (1). I wonder if the authors analyzed how often this occurs in their Preference Dataset, and whether such cases affect the model’s final performance or rationale quality.

2. The Verifier LLM seems to rely on confidence-based flagging, but model confidence can be unreliable (2). It would strengthen the work to include a comparison against a human or expert baseline to assess how trustworthy these verifications actually are.

3. METEORA’s efficiency gains partly come from using 80% fewer evidence chunks than top-k selection, but it’s unclear whether this still holds for multi-hop reasoning tasks. In such cases, reducing evidence too aggressively might hurt completeness — fewer chunks don’t necessarily mean better outcomes.

Ref:

(1) https://arxiv.org/abs/2501.07301

(2) https://cdn.openai.com/papers/simpleqa.pdf

---

> ### Author Response · Authors · 2025-11-20
> **Authors’ Response 1**
>
> We thank the reviewer RMUm for identifying our fresh perspective on re-ranking by grounding it in explicit reasoning as cleaner and more interpretable alternative to standard top-k retrieval.
>
> > **W1** : While METEORA shows clear performance gains, the paper doesn’t provide much concrete analysis or evidence to quantify its initial claims around improved interpretability and credibility. It would be helpful to see a more systematic evaluation of these aspects.
>
>
> **A1**:
> Appendix A.2 of the paper has end-to-end examples showing how the rationale trail makes the system interpretable and auditable given the sequence of rationales and the answer generated.
>
> In addition, to check the interpretability and robustness of the instructions, we conducted a human evaluation on 100 random samples across six datasets (QASPER, Contract-NLI, PrivacyQA, FinQA, CUAD, and MAUD). Four college graduate students from diverse backgrounds reviewed flagged evidence chunks alongside the flagging instructions. Annotators were kept blind from ground truth as well as from each other’s assignments.
>
> Interpretability is measured by how confident an annotator felt that a given instruction was applicable to the chunks, on an integer scale from 1 to 5. The average score was 3.64 out of 5, indicating that the instructions were judged to be interpretable and applicable by humans about 73% of the time for the given queries. An average confidence score of ≥ 3.5 (on a scale of 1 to 5) across all evaluators suggests that giving annotators access to the same instructions that RAG used to flag irrelevant or poisoned chunks helped them make better decisions.
>
> Robustness is measured by the reliability of the annotators’ flagging decisions, which achieved an average accuracy of ≈ 86% against the ground truth.
>
> We will add Experiment setup and its results in the final version of the paper.
>
> >**W2** . The main novelty seems to be the unsupervised evidence selection approach, but its real advantages aren’t clearly demonstrated — the paper lacks ablation or comparative analyses to justify its effectiveness.
>
> **A2**: METEORA builds upon the limitation from traditional re-rankers (e.g., SBERT, CrossEncoder) by replacing arbitrary top-k heuristics with rationale-conditioned selection. METEORA is a framework that solves the black box problem by replacing opaque re-ranking with explicit reasoning. As demonstrated in Figure 1, METEORA generates rationales that explain why specific evidence is relevant to a query, then uses these same rationales to verify evidence consistency and detect adversarial content. Consequently, METEORA consistently outperforms these baselines across all key metrics.
>
> To validate the effectiveness of our architecture, we present ablation studies on expansion and DPO (Table 2) and analyze specific verifier flags (Table 4). We have also added new results measuring performance on CP and Generation tasks with the Verifier completely removed to further isolate its impact as shown in Table R1 and Table R2.
>
> | Model                | QASPER             | C-NLI              | FinQA              | PrivacyQA          | CUAD               | MAUD               | Average             |
> |---|----|---|----|----|-------|-------|-----|
> | METEORA              | P: 0.26, R: 0.99   | P: 0.35, R: 1.00   | P: 0.12, R: 0.95   | P: 0.23, R: 0.98   | P: 0.12, R: 0.93   | P: 0.03, R: 0.72   | P: **0.19**, R: 0.93    |
> | METEORA w/o Verifier | P: 0.25, R: 1.00   | P: 0.34, R: 1.00   | P: 0.10, R: 0.97   | P: 0.22, R: 1.00   | P: 0.11, R: 0.94   | P: 0.02, R: 0.75   | P: 0.17, R: **0.94**    |
>
> **Table R1:** CP Task results of METEORA without Verifier. Here P = Precision, R = Recall.
> | Model                | QASPER | C-NLI | FinQA | PrivacyQA | CUAD | MAUD | Average |
> |-------|:------:|:-----:|:-----:|:------:|:----:|:----:|:-----:|
> | METEORA              |  0.74  | 0.76  | 0.71  |   0.65    | 0.53 | 0.44 |  **0.64**   |
> | METEORA w/o Verifier |  0.70  | 0.68  | 0.65  |   0.58    | 0.45 | 0.38 |  0.57   |
>
> **Table R2:** Generation Task results of METEORA without Verifier

---

> ### Author Response · Authors · 2025-11-20
> **Authors' Response 2**
>
> > **Q1**. Prior work has shown that correct answers don’t always come from correct reasoning (1). I wonder if the authors analyzed how often this occurs in their Preference Dataset, and whether such cases affect the model’s final performance or rationale quality.
>
> **A1**: The possibility of a correct answer arising from flawed evidence selection or poor reasoning is precisely what our methodology is designed to prevent in the RAG pipeline. We directly address this using Direct Preference Optimization (DPO). DPO’s primary role is to tune the LLM to prioritize the quality of the reasoning itself. This ensures that correct answers are rewarded only when supported by a robust, high-quality rationale, actively suppressing the potential for "by chance" document selections driven by incorrect reasoning.
>
> The effectiveness of this approach is validated through both our quantitative and qualitative results:
>
> 1. Quantitative Validation: Our ablation studies confirm the necessity of this approach (refer to Table 2). METEORA performs significantly better with DPO than without it, showing marked improvements of 0.93 vs 0.88 in terms of recall. This demonstrates that DPO is successfully preference-tuning the model to prioritize reliable evidence selection over chance (50%).
>
> 2. Qualitative Validation: Furthermore, our human evaluation provides crucial qualitative support. It confirmed that the DPO-tuned model generates demonstrably better rationales in terms of Interpretability (a human can clearly understand the reason for selecting evidence). This ensures the system prioritizes a strong, auditable evidence trail rather than a “fortunate” final answer.
>
> > **Q2**. The Verifier LLM seems to rely on confidence-based flagging, but model confidence can be unreliable (2). It would strengthen the work to include a comparison against a human or expert baseline to assess how trustworthy these verifications actually are.
>
> **A2**: Due to the unreliable nature of the confidence score, our Verifier LLM does not solely depend on the confidence score but uses structured, rationale based flags along with it.
> Evidence is flagged for three types of issues: (1) factual violations, when content contradicts well-established facts with high confidence (and uses threshold); (2) contradiction, when evidence is logically inconsistent with previously verified evidences; and (3) instruction violations, when evidence fails to meet criteria embedded in the rationales. Flagged evidence is discarded, and only the filtered set proceeds to generation
>
> The Verifier is designed to raise three distinct flag types: Factual Flags, Rationale-Driven Flags, and Contradiction Flags. Factual Flag depends on a high-confidence threshold and efficiently remove instances where a well-known piece of information has been poisoned. Rationale-Driven Flag is an instruction flag and is primarily driven by the rationale frame (the model's explicit reasoning) and the context of prior verified information. Contradiction Flag is used for the model to check for contradictions in the information present in the evidence chunks.
> As shown in the Table below, METEORA prioritizes instruction violations over factual and contradiction violations (taken from Table 4 of the paper). The high dominance of  instruction violations indicates that verification is primarily driven by the LLM detecting when the rationale cannot support the selected evidence.
> | Flag type                  | Contribution range |
> |----------------------------|--------------------|
> | Instruction violations          | 32–46%             |
> | Contradiction violations        | 0–2%               |
> | Factual violations              | 0–10%              |
> We will clarify the above observations in the paper.

---

> ### Author Response · Authors · 2025-11-21
> **Authors' Response 3**
>
> > **Q3**: METEORA’s efficiency gains partly come from using 80% fewer evidence chunks than top-k selection, but it’s unclear whether this still holds for multi-hop reasoning tasks. In such cases, reducing evidence too aggressively might hurt completeness — fewer chunks don’t necessarily mean better outcomes.
>
> **A3**. Our evaluation employs datasets intrinsically designed for multi-hop reasoning, including QASPER, Contract-NLI, PrivacyQA, CUAD, and MAUD [1]. In these datasets, the correct output often requires synthesizing information from multiple chunks within a document. The ability to handle multi-hop queries is entirely a function of the retriever, which selects the source documents. The re-ranker, however, simply treats all retrieved text chunks equally, ignoring which document they originally came from.
>
> Our analysis of over 20,000+ instances across these datasets confirms that 74% relied on a single evidence chunk, while the remaining 26% required multiple (multi-hop) correct chunks. We evaluated METEORA’s performance across these two groups, finding remarkable consistency. With a single correct evidence chunk, METEORA achieved a precision of 0.19 and a recall of **0.94**. For instance, demanding multiple correct evidence chunks, the performance was nearly identical, with a precision of 0.19 and a recall of **0.91**.
>
> This clearly demonstrates METEORA's effectiveness using single or multiple evidence. This success is rooted in the fact that our 80% reduction in the chunk is relative to the baseline methods. Crucially, we did not find a single instance in our entire evaluation in which METEORA under-selected the evidence; the number of selected chunks was always sufficient to include the full set of correct evidence. METEORA maintains its robust efficiency gains without sacrificing the completeness required for multi-hop tasks.
>
> **References**
>
> [1] Pipitone, N. and Alami, G.H., 2024. Legalbench-rag: A benchmark for retrieval-augmented generation in the legal domain. arXiv preprint arXiv:2408.10343.

---

> ### Comment · Reviewer_RMUm · 2025-11-22
>
> Thank you for the detailed additional clarifications and new experiments. They help address parts of my earlier concerns, but a few issues remain.
>
> While the human evaluation is appreciated, what is measured does not fully align with the original claims around interpretability. The criteria appear somewhat ad-hoc, and I still feel that a more systematic and well-defined set of metrics would be necessary to convincingly support interpretability improvements.
>
> Relatedly, the aspect of credibility is still not directly evaluated. The current experiments focus on instruction applicability and annotator agreement, which do not clearly speak to whether the system meaningfully increases users’ perceived reliability or trust.
>
> Regarding the “unsupervised” evidence selection component, I still find its specific contribution somewhat unclear. It would be helpful to more explicitly isolate and quantify how much this unsupervised mechanism contributes beyond standard supervised or heuristic re-ranking approaches.
>
> Finally, the ablation studies, while informative, show relatively modest differences. It is difficult to assess whether these gaps are significant, and whether the performance gains justify introducing a multi-stage rationale + verifier architecture of this complexity.

---

> ### Author Response · Authors · 2025-11-24
> **Response 1 out of 2**
>
> > **C1** : While the human evaluation is appreciated, what is measured does not fully align with the original claims around interpretability. The criteria appear somewhat ad-hoc, and I still feel that a more systematic and well-defined set of metrics would be necessary to convincingly support interpretability improvements.
>
> **R1** : We would like to clarify how our human evaluation aligns with a standard notion of interpretability, rather than it being ad-hoc. We will explicitly adopt the widely used definition: **“Interpretability is the degree to which a human can understand the cause of a decision.”**[1][2]
>
> In METEORA, the key decisions are which evidence chunks are flagged as poisoned/problematic and which are kept as valid evidence before generation. Our human study is designed exactly around this decision-level notion. For each query-chunk pair, annotators are given:
>
> * the query and the chunk_text,
> * the same flagging_instructions METEORA uses (plain-language rules describing when a chunk should be flagged),
> * the flags produced by the Verifier (e.g., CONTRADICTION, factual error).
>
> They are then asked a single binary question: **“If you applied these instructions, would they correctly identify this chunk as poisoned/problematic?”**
>
> Annotators answer YES/NO and provide a 1-5 confidence score, where:
> 1. very uncertain, mostly guessing
> 2. low confidence, weak cues
> 3. moderate confidence, some evidence
> 4. high confidence, strong evidence
> 5. very high confidence, compelling evidence
>
> This setup is to test: **given the same rationale frame and instructions as METEORA, can a human reproduce and justify the system’s keep/flag decision?** The ≈86% accuracy and 3.64/5 average confidence indicate that humans can reliably understand and reconstruct the cause of these evidence-level decisions. This is precisely the level of interpretability we claim: evidence decisions are understandable and reproducible by humans using the same rationale frame.
>
> Furthermore, the positive and negative end-to-end examples in the appendix show how these local decisions compose into a traceable chain (query -> rationales -> selected/flagged chunks -> answer). Taken together, the qualitative examples and the human study provide a systematic evaluation of decision-level interpretability under this definition, rather than just a set of anecdotal case studies.
>
> > **C2** : Relatedly, the aspect of credibility is still not directly evaluated. The current experiments focus on instruction applicability and annotator agreement, which do not clearly speak to whether the system meaningfully increases users’ perceived reliability or trust.
>
> **R2** : Here there is a terminological mismatch more than a gap in evidence. We do not use **“credibility”** to mean a user-interface notion of **“how much do you like this system?”** In METEORA, credibility is defined structurally:
>
> Interpretability of evidence decisions (as above: humans can understand and simulate why chunks are flagged or kept).
> Traceability of the end-to-end reasoning trail (query -> rationales -> evidence -> answer), illustrated explicitly with positive and negative cases.
>
> Robustness to poisoned or misleading chunks, shown by Verifier ablations (Tables 4, R1, R2) and by the fact that human annotators, using the same instructions, correctly identify problematic chunks with high accuracy.
>
> Our human study is therefore not asking, “Do you feel this system is trustworthy?” but **“given the same rationale frame, do you agree this chunk should be flagged, and why?”** This is the notion of credibility we target: a system whose evidence decisions are interpretable, traceable, and demonstrably robust to corruption. Measuring UI-level perceived trust is important future work, but it is distinct from the technical objective of making the evidence pipeline credible in this structural sense.
>
> **References:**
>
> [1] Miller, Tim. "Explanation in artificial intelligence: Insights from the social sciences." Artificial intelligence 267 (2019): 1-38.
>
> [2] Zachary C. Lipton. 2018. The mythos of model interpretability. Commun. ACM 61, 10 (October 2018), 36–43. https://doi.org/10.1145/3233231

---

> ### Author Response · Authors · 2025-11-24
> **Response 2 out of 2**
>
> > **C3** : Regarding the “unsupervised” evidence selection component, I still find its specific contribution somewhat unclear. It would be helpful to more explicitly isolate and quantify how much this unsupervised mechanism contributes beyond standard supervised or heuristic re-ranking approaches.
>
> **R3** : The “unsupervised” Evidence Chunk Selection Engine (ECSE) plays a concrete role in METEORA.
>
> First, it needs no labels and no extra re-ranker. ECSE does not rely on any chunk-level supervision or a fine-tuned cross-encoder. It works purely from METEORA’s own rationales and simple similarity statistics over the retrieved chunks.
>
> Second, it removes the global top-k hyperparameter. Instead of a brittle, hand-tuned top-k re-ranking step, ECSE uses an adaptive, rationale-driven selection rule to decide how many chunks to keep for each query.
>
> Our ablations already show that when ECSE is removed and we revert to a fixed top-k strategy, performance collapses toward the baseline re-ranker: we lose both robustness and efficiency. We will make this comparison more explicit in the paper. The key point is that ECSE is not a cosmetic add-on; it is what allows METEORA to:
> * avoid training a new supervised re-ranker,
> * eliminate a brittle global top-k parameter, and
> * still match or exceed baseline evidence coverage while using substantially fewer chunks.
>
> This is the core contribution of the “unsupervised” component.
>
>  > **C4** : Finally, the ablation studies, while informative, show relatively modest differences. It is difficult to assess whether these gaps are significant, and whether the performance gains justify introducing a multi-stage rationale + verifier architecture of this complexity.
>
> **R4** :  We believe this impression comes from looking at one component in isolation rather than the full picture of what METEORA optimizes: downstream generation quality under realistic evidence budgets.
>
> **Noise vs recall trade-off.**
>
> METEORA improves downstream generation by minimizing input noise rather than simply maximizing recall at any cost. In real systems, tuning an optimal k per query is impractical. We therefore evaluate baselines at METEORA’s average evidence count, which reflects how a deployment would operate. While increasing k can artificially boost recall, it also injects substantial noise into the context, which Du et al. [1] and our own experiments show directly harms answer quality. We observe this degradation both for production-grade models (e.g., GPT-4o) and for open-source LLMs. Evaluating at an artificially high “best-k” therefore optimizes the wrong objective: it helps the retrieval metric but penalizes the actual goal of the RAG pipeline, accurate generation.
>
> **Precision-Recall curves across k.**
>
> To make this explicit, we include Precision-Recall curves with k ranging from 1 to 64 (Figure [link](https://anonymous.4open.science/r/METEORA-DC46/README.md)). These curves mark, for each dataset, the k that baselines must use to match METEORA’s recall. Across datasets, baselines typically require many more chunks to reach the same recall level, i.e., METEORA achieves comparable coverage with roughly 80% less evidence. That “missing” 80% is exactly the noise that hurts generation.
>
> **Combined effect of all components.**
>
> When ECSE’s noise-minimizing selection, DPO-tuned rationales, and the Verifier are taken together, the system delivers:
> * consistently better or comparable generation quality at the same (or smaller) evidence budget,
> * high recall without over-selecting,
> * significantly reduced context size, and
> * a fully interpretable and traceable evidence pipeline.
>
> From the standpoint of high-stakes legal/financial RAG, this combination is precisely what justifies the architecture: a system that is not just marginally higher on a retrieval metric but meaningfully better at generating accurate answers with far less noise while remaining interpretable, traceable, and robust.
>
> **References:**
>
> [1] Yufeng Du, Minyang Tian, Srikanth Ronanki, Subendhu Rongali, Sravan Babu Bodapati, Aram Galstyan, Azton Wells, Roy Schwartz, Eliu A Huerta, and Hao Peng. 2025. Context Length Alone Hurts LLM Performance Despite Perfect Retrieval. In Findings of the Association for Computational Linguistics: EMNLP 2025, pages 23281–23298, Suzhou, China. Association for Computational Linguistics.

---

> ### Author Response · Authors · 2025-11-27
> **Follow-up to Reviewer RMUm**
>
> Thank you for the careful follow-up. In the final version we will (i) explicitly align our human study with standard decision-level interpretability, (ii) clarify what we mean by “credibility,” and (iii) more clearly isolate the unsupervised selector’s contribution. If these revisions address your main concerns, we would be grateful if you could consider revisiting your overall score.

---

### Official Review · Reviewer_y1nd · 2025-10-30

**Soundness:** 3
**Presentation:** 3
**Contribution:** 3
**Rating:** 8
**Confidence:** 4

**Summary:**

METEORA is a RAG system designed for sensitive domains that require interpretability and robustness. The system generates rationales (explicit reasoning) via a preference tuned LLM rather than relying on direct similarity computations between query and evidence.  Rationales are then used to guide evidence selection. The approach also includes adaptively similarity cutoffs via elbow detection, context expansion, and verification against poisoned or misleading content. Evaluations indicate that the approach achieves significant improvements over existing approaches.

**Strengths:**

1. Use of rationales to both select evidence and to explain that selection clearly to the user. Applying dpo to optimize rationale generation is also interesting.

2. The paper proposes a set of practical optimizations that could be applied to any RAG pipeline that I believe would likely lead to improvements in overall performance. This is a valuable contribution.

**Weaknesses:**

1. I see that a preference tuned LlaMA-3.1-8b was used for rationale generation and evidence verification in the experiments. Is this  LLM available for general use? I didn't see reference to it in the repo.

2. Does METEORA have an sdk that can be used to interface with the framework?

3. As I understand the rationale includes flagging instructions, I would assume that inclusion of flagging instructions may affect the quality of evidence selection? I dont see ablation study that addresses this.

**Questions:**

Is the DPO trained model available?
SDK available?

---

> ### Author Response · Authors · 2025-11-20
> **Authors’ Response**
>
> Thank you for the review and feedback.
> >**Weakness 1:** I see that a preference tuned LlaMA-3.1-8b was used for rationale generation and evidence verification in the experiments. Is this LLM available for general use? I didn't see reference to it in the repo.
>
> **A1.** We will share our preference-tuned model on Hugging Face for easy distribution and will mention this in the paper.
>
> >**Weakness 2:** Does METEORA have an sdk that can be used to interface with the framework?
>
> **A2.** METEORA currently uses modular Python components, and the anonymous GitHub link to METEORA’s code is already included in the paper. We plan to release a simple 'import meteora' style SDK in the future.
>
> >**Weakness 3:** As I understand the rationale includes flagging instructions, I would assume that inclusion of flagging instructions may affect the quality of evidence selection? I dont see ablation study that addresses this.
>
> **A3.** Yes, flagging instructions affect evidence selection. We ran a new ablation without the Verifier (that is, without flagging instructions), which resulted in a **7%** drop in accuracy, as shown in Table R2. This indicates the importance of flagging instructions. We will add this study to the final version of the paper.
>
>
>
> | Model                | QASPER             | C-NLI              | FinQA              | PrivacyQA          | CUAD               | MAUD               | Average             |
> |----------------------|--------------------|--------------------|--------------------|--------------------|--------------------|--------------------|---------------------|
> | METEORA              | P: 0.26, R: 0.99   | P: 0.35, R: 1.00   | P: 0.12, R: 0.95   | P: 0.23, R: 0.98   | P: 0.12, R: 0.93   | P: 0.03, R: 0.72   | P: **0.19**, R: 0.93    |
> | METEORA w/o Verifier | P: 0.25, R: 1.00   | P: 0.34, R: 1.00   | P: 0.10, R: 0.97   | P: 0.22, R: 1.00   | P: 0.11, R: 0.94   | P: 0.02, R: 0.75   | P: 0.17, R: **0.94**    |
>
> *Table R1: CP Task results of METEORA without Verifier (P = Precision, R = Recall)*
>
> ----
>
> | Model                | QASPER | C-NLI | FinQA | PrivacyQA | CUAD | MAUD | Average |
> |----------------------|:------:|:-----:|:-----:|:---------:|:----:|:----:|:-------:|
> | METEORA              |  0.74  | 0.76  | 0.71  |   0.65    | 0.53 | 0.44 |  **0.64**   |
> | METEORA w/o Verifier |  0.70  | 0.68  | 0.65  |   0.58    | 0.45 | 0.38 |  0.57   |
>
> *Table R2: Generation Task results of METEORA without Verifier*

---

### Note · Program_Chairs · 2026-01-17
**Submission Desk Rejected by Program Chairs**

The following references in this submission do not refer to real documents and/or have major errors in bibliographic information:

 Zsolt Karpati and Norbert Szabo. Governing black boxes: On the use of retrieval-augmented systems in law. ICAIL, 2023.
Yixuan Jia et al. Bridging the gap between retrieval and generation: Rationale-aware dense passage retrieval for open-domain question answering. In Proceedings of the 2025 Conference of the North American Chapter of the Association for Computational Linguistics: Human Language Technologies, 2025.
Arjun Verma et al. Infusion attacks in retrieval-augmented generation. arXiv preprint arXiv:2402.00789, 2024.
Linyi Yang et al. Grpo: Generalized reinsertion preference optimization for instruction tuning. arXiv preprint arXiv:2309.02654, 2023.